# Minimax Optimal Regret Bound for Reinforcement Learning with Trajectory Feedback

## Abstract

We study the reinforcement learning (RL) problem with trajectory feedback. The trajectory feedback based reinforcement learning problem, where the learner can only observe the accumulative noised reward along the trajectory, is particularly suitable for the practical scenarios where the agent suffers extensively from querying the reward in each single step. For a finite-horizon Markov Decision Process (MDP) with $S$ states, $A$ actions and a horizon length of $H$, we develop an algorithm that enjoys an optimal regret of $\tilde{O}\left(\sqrt{SAH^3K}\right)$ in $K$ episodes for sufficiently large $K$.[1] To achieve this, our technical contributions are two-fold: (1) we incorporate reinforcement learning with linear bandits problem to construct a tighter confidence region for the reward function; (2) we construct a reference transition model to better guide the exploration process.

## 1 Introduction

In the standard reinforcement learning (RL) formulation, it is assumed that the agent acts in an unknown environment, and in each step, the agent receives feedback in the form of a state-action dependent reward signal, and then transits to the next state. Although such an interaction model might be reasonable when a simulator is available, for real-life applications, such reward feedback model could be hard to realize. For practical scenarios, querying the reward function could be costly, or even impossible in certain circumstances.

As a motivating example, in healthcare, a doctor repeatedly interacts with a patient for the purpose of treatment. In each step, the doctor decides an action (e.g., taking some medicine) and observes the new state (including information like body temperature or blood pressure). On the other hand, the state-action dependent reward signal could be costly to observe, since the extent to which the disease has been cured might be expensive to measure as it requires comprehensive medical tests. In this case, in order to apply the RL framework, it is more reasonable to assume that in each step, the agent observes only the current state, and the cumulative reward value is revealed only after a whole trajectory is finished.

As another example, in autonomous car driving, defining a state-action dependent reward function could be a challenging task, as it requires associating all possible state-action pairs with a real number. A possible workaround is to have human experts involved to produce the reward signals. However, defining reward signals could be a highly subjective matter, and waiting for reward values from human experts could take unacceptable amount of time from the perspective an a RL algorithm.

To circumvent issues mentioned above, practitioners often rely on heuristics (e.g., reward shaping (Ng et al., 1999) or reward hacking (Amodei et al., 2016)). RL with trajectory feedback has been recently proposed in Efroni et al. (2021) as a more principled framework to the deal with the aforementioned issues. In this framework, the agent no longer has access to a per state-action reward function. Instead, it receives the cumulative reward on the trajectory as well as all the visited state-action pairs on the trajectory. Clearly, this new feedback model is weaker than the standard RL setting and could be more applicable for real-life scenarios. In Efroni et al. (2021), new algorithms

---

[1]Throughout this paper, we use $\tilde{O}(\cdot)$ to suppress logarithmic factors.

based on the principle of optimism and Thompson sampling were proposed. Although all these algorithms achieve $\sqrt{K}$-type regret bounds, the dependence on the number of state-action pairs is far from being optimal. Obtaining nearly optimal regret bounds in this setting is the main focus of the present paper.

**Our Contribution.** In this paper, we prove a minimax optimal regret with trajectory feedback for sufficiently large $K$. Formally, we present the result as follows.

**Theorem 1** (Informal version of Theorem 7). *Fix $\delta > 0$. For any episodic MDP with trajectory feedback, there exists an algorithm (Algorithm 1) such that with probability $1 - \delta$, the regret in $K$ episodes does not exceeds $\tilde{O}\left(\sqrt{SAH^3K}\right)$ for sufficiently large $K$. Here $S$ is the number of states, $A$ is the number of actions, $H$ is the horizon length, and $K$ is the total number of episodes.*

It it known that even for episodic MDPs, even if the agent has access to the per state-action reward function, the regret bound of any RL algorithm is lower bounded by $\Omega(\sqrt{SAH^3K})$ (Jin et al., 2018; Domingues et al., 2021) [2]. Thus, the leading term of the regret bound in Theorem 1 has near-optimal dependence on the number of states $S$ and horizon length $H$, and therefore, our regret bound is asymptotically nearly optimal.

Conceptually, Theorem 1 shows that RL with trajectory feedback, a seemingly harder setting, has the same asymptotically optimal regret bound as the standard RL setting. Therefore, at least statistically, RL with trajectory feedback is no harder than the standard setting.

On the other hand, the algorithm for achieving Theorem 1 is not computationally efficient as it requires maintaining a set of deterministic policies during its execution, and an intriguing open problem is to design computationally-efficient algorithms RL with trajectory feedback with asymptotically nearly optimal bounds, or showing that such an algorithm does not exist.

The remaining part of this paper is organized as follows. Section 2 give an overview of related work. Section 3 introduces necessary technical backgrounds and notations. Section 4 gives an overview of the technical challenges for obtaining our new results and their solutions. Section 5 introduces the formal definition of our algorithms together with an overview of is analysis. Most of the proofs are deferred to the supplementary material.

## 2 RELATED WORK

**RL with Limited Feedback.** As mentioned in the introduction, RL with trajectory feedback was introduced in Efroni et al. (2021). Cohen et al. (2021) provided an algorithm that works for RL with trajectory feedback even when the noise is adversarially chosen. Chatterji et al. (2021) considered a more general setting where the reward revealed to the learner is no longer the cumulative reward on the sampled trajectory, but instead drawn from a logistic model. It is an interesting future direction to generalize our techniques to their setting and obtain nearly optimal regret bounds.

Very recently, Cassel et al. (2024) considered RL with trajectory feedback in linear MDPs (Yang & Wang, 2019; Jin et al., 2020) and achieved a regret bound of $\tilde{O}(\sqrt{d^5H^7K})$. Translating their regret bound to the tabular setting considered in the present paper, the regret bound would be $\tilde{O}(\sqrt{S^5A^5H^7K})$ which is far from being asymptotically nearly optimal. It would be interesting to generalize our techniques to RL with trajectory feedback when function approximation schemes are used and obtain improved regret bounds.

Preference-based RL (PbRL) is another RL paradigm to deal with the lack of a reward function in various real-world scenarios. We refer interested readers to Wirth et al. (2017) for an overview of PbRL. Theoretical results for PbRL have been obtained in the tabular setting (Novoseller et al., 2020; Xu et al., 2020b; Saha et al., 2023) and various function approximation settings (Chen et al., 2022; Wu & Sun, 2023; Wang et al., 2023). Preference-based learning has also been studied in bandit setting under the notion of "dueling bandits" (Yue et al., 2012; Falahatgar et al., 2017; Bengs

---

[2]In fact, the regret lower bound proved by Jin et al. (2018) is $\Omega(\sqrt{SAH^2T})$ with $T = KH$, which would be translated to $\Omega(\sqrt{SAH^3K})$ using our notations.

et al., 2021; Xu et al., 2020a). Dueling bandits can be thought as a special case of PbRL with a single state and horizon length $H = 1$.

**Linear Bandits.** Linear bandits is a classical setting for modeling sequential decision-making problems, and various sample complexity bounds and regret bounds have obtained in this setting and its generalizations (Dani et al., 2008; Abbasi-Yadkori et al., 2011; Li et al., 2019; Filippi et al., 2010; Li et al., 2019). We refer readers to Lattimore & Szepesvári (2020) for a comprehensive survey on this topic. As observed in Efroni et al. (2021), there is a deep connection between RL with trajectory feedback and linear bandits. More specifically, RL with trajectory feedback can be understood as an instance of linear bandits over a convex set. Such a connection is also exploited in the present paper which will be discussed in more details in Section 4.

**Regret Bounds for the Standard RL Setting.** There is a long line of work studying regret minimization in RL (Kakade, 2003; Jaksch et al., 2010; Azar et al., 2017; Jin et al., 2018; Zanette & Brunskill, 2019; Zhang & Ji, 2019; Zhang et al., 2020; 2022b; 2024). In particular, an asymptotically nearly optimal regret upper bound of $\tilde{O}\left(\sqrt{SAH^3K}\right)$ has been known in the literature (Azar et al., 2017), and more recent work typically focuses on the lower order terms, i.e., by considering the case where the total number of episodes $K$ is not that large compared to the number of states $S$, the number of actions $A$ and the horizon length $H$. In particular, the most recent work by Zhang et al. (2024) shows that an upper bound of $\tilde{O}\left(\sqrt{SAH^3K} + KH\right)$ can be achieved for any $K \geq 1$.

Notably, in order to learn the transition model, in this paper we use an algorithmic framework based on policy elimination similar to that used in Zhang et al. (2022b), although the algorithm in Zhang et al. (2022b) is designed for the standard RL setting which does not require the tighter confidence region construction for reward functions which is the main technical contribution of the present paper.

## 3 PRELIMINARIES

**Episodic reinforcement learning with trajectory feedback.** An MDP is defined as $M = \langle \mathcal{S}, \mathcal{A}, R, P, \mu \rangle$, where $\mathcal{S}$ is the state space, $\mathcal{A}$ is the action space, $R = \{\mathcal{R}_h(s, a)\}_{(s,a)\in\mathcal{S}\times\mathcal{A}, h\in[H]}$ is the unknown reward distribution, $P = \{P_{h,s,a}\}_{(s,a)\in\mathcal{S}\times\mathcal{A}, h\in[H]}$ is the unknown transition model and $\mu$ is the initial distribution. We assume that the reward distribution $\mathcal{R}_h(s, a)$ is supported by $[0, 1]$ for any $(h, s, a)$ with mean $R_h(s, a)$. In each episode, the agent starts at $s_1$, which is drawn according to $\mu$. It then proceeds to take actions, transitioning to the next state step by step, finally constructing the trajectory $\{(s_h, a_h, s_{h+1})\}_{h=1}^H$. In the end of the episode, the agent receives a trajectory reward feedback $Y = \sum_{h=1}^H r_h(s_h, a_h)$, where each $r_h(s_h, a_{,h})$ is independently drawn according to $\mathcal{R}_h(s_h, a_h)$.

A (deterministic) policy $\pi$ can be viewed as a collection of mappings $\{\pi_h\}_{h=1}^H$ where each $\pi_h : \mathcal{S} \to \mathcal{A}$ is a map from the state space to the action space. Let $\mathcal{T}$ denote the set of all trajectories and $\Pi_{\det}$ denote the set of all deterministic policies. In our algorithm, we also consider mixtures of deterministic policies. More specifically, a mixture of deterministic policies $\overline{\pi}$ could be regarded as a distribution over $\Pi_{\det}$.

Given a policy $\pi$, the (optimal) $Q$-function and value function are given by[3]

$$Q_h^\pi(s, a) = \mathbb{E}_\pi\left[\sum_{h'=h}^H r_{h'}(s_{h'}, a_{h'})\Big|(s_h, a_h) = (s, a)\right]; \qquad Q_h^*(s, a) = \sup_{\pi\in\Pi_{\det}} Q_h^\pi(s, a);$$

$$V_h^\pi(s) = \mathbb{E}_\pi\left[\sum_{h'=h}^H r_{h'}(s_{h'}, a_{h'})\Big|s_h = s\right]; \qquad V_h^*(s) = \max_a Q_h^*(s, a).$$

Let $\pi^*$ be an optimal policy such that $Q_h^*(s, a) = Q_h^{\pi^*}(s, a)$ for all $(s, a, h)$.

---

[3]It is well known that optimal $Q$(value) function could be reached by a deterministic policy.

Define $W^\pi(r, p) := \mathbb{E}_{\pi, p, s_1 \sim \mu} \left[ \sum_{h=1}^H r_h(s_h, a_h) \right]$ and $W^*(r, p) = \max_{\pi \in \Pi_{\det}} W^\pi(r, p)$. Let $\pi^k$ denote the policy in the $k$-th episode. Then the regret is given by

$$\text{Regret}(K) := \sum_{k=1}^K \left( W^*(R, P) - W^{\pi^k}(R, P) \right). \tag{1}$$

**Notations.** In this paper, we use $\mathbb{E}_{\pi, p}[\cdot]$ ($\Pr_{\pi, p}[\cdot]$) to denote the expectation (probability) under the policy $\pi$ and transition probability $p$. In particular, $\Pr_{\pi, P}[\tau] = \Pi_{h=1}^H (\mathbb{I}[\pi_h(s_h) = a_h] P_{s_h, a_h, h, s_{h+1}})$ is the probability of $\tau = \{(s_h, a_h)\}_{h=1}^H$ under $(\pi, p)$. We also define the general occupancy function $d_p^\pi(s, a, h) = \mathbb{E}_{\pi, p} [\mathbb{I}[(s_h, a_h) = (s, a)]]$. We use $d_p^\pi$ to denote the $SAH$-dimensional vector $\{d_p^\pi(s, a, h)\}_{(s, a, h) \in \mathcal{S} \times \mathcal{A} \times [H]}$. Similarly, we may also regard $R$ as a $SAH$-dimensional vector $\{R_h(s, a)\}_{(s, a, h) \in \mathcal{S} \times \mathcal{A} \times [H]}$. For $N \geq 1$, we use $[N]$ to denote the set $[1, 2, \ldots, N]$. Given a trajectory $\tau = \{(s_h, a_h)\}_{h=1}^H$, we let $\phi_\tau \in \mathbb{R}^{SAH}$ to be the vector such that $\phi_\tau(s', a', h) := \mathbb{I}[(s', a') = (s_h, a_h)]$. We use $\mathbf{I}$ to denote the $SAH$-dimensional identity matrix. For two vector $x, y$ with the same dimension, we write $x^\top y$ as $xy$ for simplicity. For $p \in \Delta^S$ and $v \in \mathbb{R}^S$, we define the variance function as $\mathbb{V}(p, v) = pv^2 - (pv)^2$. We use $\mathcal{E}^C$ to denote the complement of the set $\mathcal{E}$.

## 4 TECHNICAL OVERVIEW

In this section, we give an overview of the technical challenges associated with obtaining the minimax optimal regret bound for RL with trajectory feedback, together with our approaches to tackle these challenges. To explain the high-level ideas, we first consider the simpler setting that the transition model $P$ is known to the algorithm, and then switch to the general setting in which case the transition model is unknown.

**Connection with Linear Bandits.** As observed in prior work on RL with trajectory feedback (Efroni et al., 2021), when the transition model, RL with trajectory feedback can be seen as an instance of linear bandits. More specifically, in each round, suppose the trajectory sampled by the agent is $\tau$, the expected trajectory reward feedback would be $\phi_\tau^\top R$, i.e., a linear function with respect to $\phi_\tau$. Based on this observation, Efroni et al. (2021) showed how to build appropriate confidence regions for RL with trajectory feedback by adapting analysis for linear bandits algorithms, and obtained a regret bound of $\tilde{O}\left(\sqrt{S^2 A^2 H^4 K}\right)$. Although it is plausible to improve their regret bound to $\tilde{O}\left(\sqrt{S^2 A H^3 K}\right)$ by a more refined analysis, it is unclear how to improve the order of $S$ in their regret bound. Indeed, in the work of (Efroni et al., 2021), RL with trajectory feedback is naïvely treated as an instance of linear bandits with feature dimension $d = SAH$, and the best known regret bound for any linear bandits algorithm is $\tilde{O}(d\sqrt{T})$ (Dani et al., 2008), or $O(\sqrt{dT \log K})$ for linear bandits with $K$ arms (Bubeck et al., 2012). Since there are $A^{SH}$ policies for an MDP, and each of them can be seen as an arm in the linear bandits problem instance, improving the order of $S$ in the regret bound of prior work requires fundamentally new ideas.

**Tighter Confidence Region Based on Trajectories.** In order to achieve a minimax optimal regret bound, our first new idea is to build a tighter confidence region by exploiting structures of the linear bandits instance associated with RL with trajectory feedback. Before getting into more details, we first review least squares regression (LSR), an estimator commonly used in linear bandits algorithms (also in prior work on RL with trajectory feedback (Efroni et al., 2021)) based on the principle of optimism in the face of uncertainty.

Given a set of data points $\{\pi^t, \tau^t, Y^t\}_{t=1}^T$, where for each $1 \leq t \leq T$, where $\pi^t$ the policy used in the $t$-th round, $\tau^t$ is the trajectory sampled by executing $\pi^t$ and the $Y^t$ is the trajectory reward feedback. Clearly, $\mathbb{E}[Y_t] = \phi_{\tau^t}^\top R$, which motivates the design of the the LSR estimator

$$\hat{R} = \arg\min_r \sum_{t=1}^T \left( Y^t - \phi_{\tau^t}^\top r \right) + \lambda \|r\|_2^2 = \Lambda^{-1} \sum_{t=1}^T \phi_{\tau^t} Y^t, \tag{2}$$

where $\Lambda = \lambda \mathbf{I} + \sum_{t=1}^T \phi_{\tau^t} \phi_{\tau^t}^\top$ is the information matrix. Optimism-based linear bandits algorithms typically maintain a set of arms, and eliminate arms outside the confidence region during the exe-

cution of the algorithm. For RL with trajectory feedback, each arm in the linear bandits instance corresponds to a deterministic policy in the original MDP.

Our construction of the tighter confidence region is based on the following two key observations:

- Although the total number of deterministic policies could be as large as $A^{SH}$, the number of trajectories is $|\mathcal{T}| = (SA)^H$;

- For any deterministic policy $\pi$, $d_P^\pi = \sum_{\tau \in \mathcal{T}} \Pr_{\pi,P}[\tau] \cdot \phi_\tau$ is a convex combination of $\{\phi_\tau\}_{\tau \in \mathcal{T}}$.

Based on these observations, instead of building confidence region for $|(d_P^\pi)^\top(\hat{R} - R)|$ for each deterministic policy $\pi$ and applying a union bound over all policies which result in suboptimal regret bounds, we consider the following event

$$\mathcal{E} := \left\{ \left| \phi_\tau^\top(\hat{R} - R) \right| \le C \left( \min \left\{ \sqrt{\phi_\tau^\top \Lambda^{-1} \phi_\tau \sigma^2 \log(2|\mathcal{T}|/\delta)}, H \right\} \right), \forall \tau \in \mathcal{T} \right\}, \quad (3)$$

where $C$ some proper constant, and $\sigma^2 \le H$ is a constant such that $\{Y^t - \phi_{\tau^t}^\top R\}_{t=1}^T$ is a group of independent zero-mean $\sigma^2$-subgaussian random variables. By standard concentration arguments, $\mathcal{E}$ holds with probability at least $1 - \delta$. We assume $\mathcal{E}$ holds in the remaining part of the discussion.

Note that second observation states that $d_P^\pi = \sum_{\tau \in \mathcal{T}} \Pr_{\pi,P}[\tau] \cdot \phi_\tau$, which implies that

$$\left| (d_P^\pi)^\top (\hat{R} - R) \right| \le \sum_{\tau \in \mathcal{T}} \Pr_{\pi,P}[\tau] \left| \phi_\tau^\top (\hat{R} - R) \right|$$

$$\le O \left( \sum_{\tau \in \mathcal{T}} \Pr_{\pi,P}[\tau] \min \left\{ \sqrt{\phi_\tau^\top \Lambda^{-1} \phi_\tau H \log(2|\mathcal{T}|/\delta)}, H \right\} \right)$$

$$\le \tilde{O} \left( H \sqrt{\sum_{\tau \in \mathcal{T}} \Pr_{\pi,P}[\tau] \min \left\{ \phi_\tau^\top \Lambda^{-1} \phi_\tau, 1 \right\}} \right) \quad (4)$$

for any policy $\pi$, where the last step holds by Cauchy-Schwarz inequality and the fact that $|\mathcal{T}| = (SA)^H$.

**Exploration by Optimal Design.** During the execution of the algorithm, we maintain a set of remaining deterministic policies $\Pi$. According to equation 4, in order to prove a uniform upper bound for $\left| (d_P^\pi)^\top(\hat{R} - R) \right|$ fro all $\pi \in \Pi$, it suffices to bound

$$\max_{\pi \in \Pi} \sum_{\tau \in \mathcal{T}} \Pr_{\pi,P}[\tau] \min \left\{ \phi_\tau^\top \Lambda^{-1} \phi_\tau, 1 \right\}. \quad (5)$$

For this purpose, we need to carefully choose a set of policies $\{\pi^t\}_{t=1}^T$, so that the quantity in equation 5 is upper bounded. As another new technical ingredient, we show how to generalize the Kiefer–Wolfowitz Theorem to our setting. In particular, in Lemma 8 in the supplementary material, we show that there exists $\bar{\pi}$ which is a mixture of deterministic policies, such that

$$\max_{\pi \in \Pi} \sum_{\tau \in \mathcal{T}} \Pr_{\pi,P}[\tau] \phi_\tau^\top \Lambda_{\bar{\pi}}^{-1} \phi_\tau = SAH, \quad (6)$$

where $\Lambda_{\bar{\pi}} := \sum_{\tau \in \mathcal{T}} \Pr_{\bar{\pi},P}[\tau] \phi_\tau \phi_\tau^\top$. Therefore, by running $\bar{\pi}$ for $T$ steps, we could collect an information matrix $\Lambda \succcurlyeq cT\Lambda_{\bar{\pi}}$ with high probability for some constant $c > 0$. Combining equation 4 and equation 6, we obtain that

$$\max_{\pi \in \Pi} \left| (d_P^\pi)^\top (\hat{R} - R) \right| \le \tilde{O} \left( H \sqrt{SAH/T} \right). \quad (7)$$

In summary, with the arguments above, for any policy set $\Pi$, we are able to collect a dataset $\{\pi^t, \tau^t, Y^t\}_{t=1}^T$ in $T$ episodes to obtain $\hat{R}$, such that

$$\max_{\pi \in \Pi} \left| W^\pi(\hat{R}, P) - W^\pi(R, P) \right| = \max_{\pi \in \Pi} \left| (d_P^\pi)^\top (\hat{R} - R) \right| \le \tilde{O} \left( \sqrt{SAH^3/T} \right). \quad (8)$$

**Online Batch Learning by Policy Elimination.** Finally, we show how to combine the two technical ingredients mentioned into the framework of online policy elimination. In this framework, the learning process is divided into consecutive batches. The algorithm maintains a policy set during its execution. Suppose the policy set maintained is $\Pi_\ell$ at the beginning of the $\ell$-th batch. The algorithm will eliminate a subset of policies from $\Pi_\ell$ to form $\Pi_{\ell+1}$ in the $\ell$-th batch. Initially, we set $\Pi_1$ to be the set of all deterministic policies. Then there will be a total of $O(\log \log K)$ batches for the whole algorithm, and there are $K_\ell = 2K^{1-\frac{1}{2^\ell}}$ episodes in the $\ell$-th batch.

As an invariant, during the execution of the algorithm, we always have that the optimal policy $\pi^* \in \Pi_\ell$ for all $\ell$. By equation 8, for each $\ell$, we obtain a set of estimated reward values $\hat{R}^\ell$ such that

$$\max_{\pi \in \Pi_\ell} \left| W^\pi(\hat{R}, P) - W^\pi(R, P) \right| \leq \tilde{O}\left( \sqrt{SAH^3/K_\ell} \right). \tag{9}$$

By setting

$$\Pi_{\ell+1} = \left\{ \pi \in \Pi_\ell : \max_{\pi' \in \Pi_\ell} W^{\pi'}(\hat{R}, P) - W^\pi(\hat{R}, P) \leq \epsilon_\ell \right\} \tag{10}$$

where $\epsilon_\ell = \tilde{O}\left( \sqrt{SAH^3/K_\ell} \right)$, it holds that $\pi^* \in \Pi_{\ell+1}$ and

$$W^*(R, P) - W^\pi(R, P) \leq \tilde{O}\left( \sqrt{SAH^3/K_\ell} \right)$$

for any $\pi \in \Pi_{\ell+1}$. Therefore, the regret in the $(\ell+1)$-th batch is bounded by

$$\tilde{O}(K_{\ell+1}\sqrt{SAH^3/K_\ell}) = \tilde{O}(\sqrt{SAH^3K}),$$

which means that the total regret is at most $\tilde{O}(\sqrt{SAH^3K})$.

**Dealing with Unknown Transition Models.** In the discussion above, we assume the knowledge of transition model $P$. Now we discuss how to remove such an assumption by learning the transition model in an online fashion. In order to implement the elimination-based online batch learning process mentioned above, we only need the transition model (i) to design the exploration policy so that equation 6 is ensured and (ii) to ensure the policy elimination step in equation 10 can be accurately implemented.

To achieve (i) and (ii), we first obtain a reference transition model $\tilde{P}$. Following the regret analysis for online batch learning in Zhang et al. (2022b), the regret stemming from learning $\tilde{P}$ can be bounded by $\tilde{O}(\sqrt{SAH^3K})$ (with lower order terms ignored). Moreover, for (i), an exact solution for equation 6 is not necessary. Instead, an approximate solution with a constant competitive ratio is sufficient to guide the exploration process, which could be found with the assistance of a reference model.

Given such an approximate transition kernel, Zhang et al. (2022b) achieves computationally efficient batch learning on the benefit of reward knowledge. In contrast, even with complete knowledge of the transition model, we suffer from inefficiency due to lack of reward information. Since we view the learning problem as a linear bandit problem with exponentially many arms, one crucial point to reaching an efficient implementation is to understand the inner structure of the arm set. In our algorithm, an important optimization problem is $\max_\pi \mathbb{E}_\pi \left[ \sum_{s,s',h,h'} \mathbb{I}[s_h = s, s_{h'} = s']r(s, s', h, h') \right]$ with fixed double-state reward function $\{r(s, s', h, h')\}$ (see line 4 in Algorithm 3). However, no existing algorithms could solve this problem (with approximation) efficiently under known transition. We leave this problem as an interesting future direction.

## 5 Algorithm

In this section, we present our algorithms. The detailed parameter settings could be found in Appendix A. The main algorithm (Algorithm 1) comprises two stages.

The first stage (line 3 in Algorithm 1) serves to acquire a coarse approximation $p$ of the transition model $P$, guiding the design of exploration policy. Instead of approximating $P$ with respect to $L_1$-norm error, we expect that the trajectory distribution under $P$ could be covered by that under $p$ up to a constant ratio. Formally, we have the definition below to measure similarity between two transition models.

**Definition 2.** *For two transition models $p$ and $p'$, we say $p$ is an $(n, x)$-approximation of $p'$ with respect to $\Pi$ iff $\mathcal{S} \times \mathcal{A} \times \mathcal{S} \times [H]$ could be divided into two sets $\mathcal{K}$ and $\mathcal{K}^{\mathrm{C}}$ such that*

$$\exp(-\log(n)/H)p'_{s,a,h,s'} \leq p_{s,a,h,s'} \leq \exp(\log(n)/H)p'_{s,a,h,s'}, \forall (s,a,h,s') \in \mathcal{K}; \tag{11}$$

$$\mathrm{Pr}_{\pi,p}[\mathcal{K}^{\mathrm{C}}] = 0, \quad \forall \pi \in \Pi_{\mathrm{det}}; \tag{12}$$

$$\max_{\pi \in \Pi} \mathrm{Pr}_{\pi,p'}[\mathcal{K}^{\mathrm{C}}] \leq x, \tag{13}$$

*where $\mathrm{Pr}_{\pi,q}[\mathcal{K}^{\mathrm{C}}]$ denote the probability of visiting $\mathcal{K}^{\mathrm{C}}$ under policy $\pi$ and transition $q$.*

The second stage consists of several consecutive batches. In each batch of the second stage (line 5 in Algorithm 1), we search for an approximate solution $\bar{\pi}$ to the design problem equation 15 given $p$ as a desired approximation of $P$. Subsequently, we execute $\bar{\pi}$ to collect the trajectory feedback, and construct reward confidence region $\mathcal{R}$ with least square regression. With the reward estimator $\hat{R}$ in hand, we then calculate the confidence region for each survived policy and proceed with policy elimination based on these computations.

---

**Algorithm 1**

1: **Input:** total number of episodes $K$.
2: **Initialization:** Set $K_0, L, K_\ell \geq 1, \epsilon_0, \sigma_0, \kappa$ according to Section A;
3: $\{\tilde{P}, \Pi_1\} \leftarrow \texttt{Ref-Model}(K_0, K)$
4: **for** $\ell = 1, 2, \ldots, L$ **do**
5: $\quad \Pi_{\ell+1} \leftarrow \texttt{Traj-Learning}(\tilde{P}, K_\ell, \Pi_\ell)$;
6: **end for**

---

### 5.1 LEARNING THE REFERENCE MODEL

We present the algorithm to learn the reference model in Algorithm 2. The algorithm consists of four distinct stages. Initially, the objective is to acquire a coarse reference model. In the subsequent stage, the focus shifts to learning a coarse reward estimator. The third stage involves gathering samples to execute policy elimination, ensuring that the remaining policies are approximately $O(\epsilon_0)$-optimal. In the final stage, we invoke `Raw-Exploration` with a larger length to obtain a more refined reference model.

**Raw exploration.** In Algorithm 2, we invoke `Raw-Exploration` (see Algorithm 6 in Appendix C) to learn a proper reference model. This algorithm is based on Algorithm 2 Zhang et al. (2022b), with slight modification so that it could be applied to general policy set $\Pi$.

---

**Algorithm 2** `Ref-Model`$(K_0, K)$

1: **Input:** length $K_0$, total length $K$;
2: **Initialization:** $\bar{K}_1 = \bar{K}_2 = \bar{K}_3 = 1000\sqrt{SAHK}, \bar{K}_4 = K_0 - 3\bar{K}_1$ ;
3: $\hat{P}_1 \leftarrow \texttt{Raw-Exploration}(\Pi_{\mathrm{det}}, \bar{K}_1)$
4: $\hat{R} \leftarrow \texttt{Reward-Regression}(P_1, \Pi_{\mathrm{det}}, \bar{K}_2)$;
5: $\Pi_1 \leftarrow \texttt{Plan}(\hat{R}, \hat{P}_1, \bar{K}_3, \Pi_{\mathrm{det}}, \epsilon_0)$;
6: $\hat{P}_2 \leftarrow \texttt{Raw-Exploration}(\Pi_1, \bar{K}_4)$;
7: **return:** $\{\hat{P}_2, \Pi_1\}$.

---

The following lemma describes the accuracy of the learned model.

**Lemma 3.** *By running `Ref-Model`$(K_0, K)$, with probability $1 - \delta$, it holds that*

- $\hat{P}_2$ *is an $(3, \sigma_0)$-approximation of $P$ with respect to $\Pi_1$;*

- $\pi^* \in \Pi_1$;

- $W^\pi(R, P) \geq W^*(R, P) - 2\epsilon_0$ *for any* $\pi \in \Pi_1$.

The proof of Lemma 3 is postponed to Appendix D.1

## 5.2 ONLINE LEARNING WITH REWARD REGRESSION

---
**Algorithm 3** Traj-Learning$(p, T, \Pi)$
---
1: **Input:** reference model $p$, length $T$, policy set $\Pi$;
2: $\hat{R} \leftarrow$ Reward-Regression$(p, T, \Pi)$;
3: $\Pi_{\text{next}} \leftarrow$ Plan$(\hat{R}, p, T, \Pi, \kappa)$;
4: **return:** $\Pi_{\text{next}}$.
---

**Reward regression.** We compute the optimal design policy according to the reference model $p$, and then collect trajectory feedback to learn the reward function. It is worth noting that the least square regression estimator $\bar{R}$ (see line 12 in Algorithm 4) might escape $[0, 1]^{SAH}$, where we construct a reward confidence region $\mathcal{R}$ (see line 13 in Algorithm 4) instead. For Algorithm 4, we have that

---
**Algorithm 4** Reward-Regression$(p, T, \Pi)$
---
1: **Input:** reference model $p$, length $T$, policy set $\Pi$;
2: **Initialization:** $\lambda \leftarrow \frac{1}{SAH^2T}$, $\Lambda \leftarrow \lambda \mathbf{I}$ , $T_1 \leftarrow \frac{T}{54 \log(2d/\delta)}$
3: **for** $t = 1, 2, \ldots, T_1$ **do**
4: $\quad \pi^t \leftarrow \max_{\pi \in \Pi} \sum_{\tau \in \mathcal{T}} \Pr_{\pi,p}[\tau] \cdot \min\{\phi_\tau^\top \Lambda^{-1} \phi_\tau, 1\}$;
5: $\quad \Lambda \leftarrow \Lambda + \sum_{\tau \in \mathcal{T}} \Pr_{\pi^t,p}[\tau] \phi_\tau \phi_\tau^\top \cdot \frac{1}{\max\{\phi_\tau^\top \Lambda^{-1} \phi_\tau, 1\}}$;
6: **end for**
7: $\bar{\pi}$ be the mixed policy which plays $\pi^t$ with probability $1/T_1$ for each $1 \leq t \leq T_1$;
8: **for** $t = 1, 2, \ldots, T$ **do**
9: $\quad$ Run $\bar{\pi}$ to get trajectory $\tau^t$ and reward $Y^t$;
10: **end for**
11: $\hat{\Lambda} \leftarrow 18\lambda \log(2d/\delta)\mathbf{I} + \sum_{t=1}^T \phi_{\tau^t} \phi_{\tau^t}^\top$;
12: $\bar{R} \leftarrow \bar{\Lambda}^{-1} \sum_{t=1}^T Y^t \phi_{\tau^t}$
13: $\mathcal{R} \leftarrow \{\tilde{R} \in [0, 1]^{SAH} : |\phi_\tau^\top \tilde{R} - \phi_\tau^\top \bar{R}| \leq 8\sqrt{H^2 \log(SAH) \log(4/\delta)\phi_\tau^\top \hat{\Lambda}^{-1} \phi_\tau}, \forall \tau\}$;
14: **if** $\mathcal{R} \neq \emptyset$ **then**
15: $\quad$ Choose $\hat{R} \in \mathcal{R}$ ;
16: **else**
17: $\quad \hat{R} \leftarrow \mathbf{0}$;
18: **end if**
19: **return:** $\hat{R}$
---

**Lemma 4.** *Assume $p$ in an $(3, x)$-approximation of $P$ with respect to $\Pi$ for some $x \geq 0$. With probability $1 - \delta$, it holds that*

$$\max_{\pi \in \Pi} \left| W^\pi(\hat{R}, P) - W^\pi(R, P) \right| \leq H\sqrt{\log(SAH)\log(4/\delta)} \cdot \left( x + 325\sqrt{\frac{SAH \log(T) \log(\frac{2SAH}{\delta})}{T}} \right),$$

*where $\hat{R} =$ Reward-Regression$(p, T, \Pi)$*

**Online batch learning.** With the reward estimator $\hat{R}$ in hand, we proceed to construct the confidence region to facilitate policy elimination. As described in Algorithm 5, for every batch, we employ reward-zero exploration to seek out the policy with nearly optimal coverage. Utilizing this policy, we can establish a uniform bound for the length of confidence intervals across all surviving policies. Formally, we have the uniform convergence result for Algorithm 5 as follows.

**Lemma 5.** *Assume that*

- $\pi^* \in \Pi$;

- *$p$ is an $(3, x)$-approximation of $P$ with respect to $\Pi$ for some $x \geq 0$;*

- *$W^\pi(u, P) \geq W^*(u, P) - y$ for some $y \geq 0$ and any $\pi \in \Pi$;*

- $\max_{\pi \in \Pi} |W^\pi(u, P) - W^\pi(R, P)| \leq z$;

- *$\epsilon \geq 2(b + z)$, where*

$$b := 30\sqrt{\frac{SAH^2(H + Sy) \log\left(\frac{8SAH}{\delta}\right)}{T}} + \frac{360S^2AH^3 \log\left(\frac{8SAH}{\delta}\right)}{T} + 4SAH^2 x.$$

*Let $\Pi_{\text{next}} = Plan(u, p, T, \Pi, \epsilon)$. With probability $1 - \delta$, it holds that:*

- *the optimal policy $\pi^* \in \Pi_{\text{next}}$;*

- *$W^\pi(R, P) \geq W^*(R, P) - 2\epsilon$ for any $\pi \in \Pi_{\text{next}}$.*

---

**Algorithm 5** $Plan(u, p, T, \Pi, \epsilon)$

---

1: **Input:** reward function $u$, transition model $p$, length $T$, policy set $\Pi$, threshold $\epsilon$
2: $\bar{\pi} \leftarrow \text{Design}(\Pi, p)$;
3: $c(s, a, h) \leftarrow \mathbb{E}_{\bar{\pi}, p}[\mathbb{I}[(s_h, a_h) = (s, a)]]$ for all $(s, a, h)$;
4: Execute $\bar{\pi}$ in the next $T$ episodes, and collect the samples as $\mathcal{D}$;
5: $N_h(s, a) \leftarrow$ the count of $(s, a, h)$ in $\mathcal{D}$;
6: **for** $(s, a, h) \in \mathcal{S} \times \mathcal{A} \times [H]$ **do**
7: $\quad \hat{p}_{h,s,a} \leftarrow$ the empirical transition probability of the samples of $(s, a, h)$ in $\mathcal{D}$;
8: **end for**
9: $\Pi_{\text{next}} \leftarrow \left\{ \pi \in \Pi : W^\pi(u, \hat{p}) \geq \max_{\pi' \in \Pi} W^{\pi'}(u, \hat{p}) - \epsilon \right\}$
10: **return:** $\Pi_{\text{next}}$.
11: **Function:** $\text{Design}(\Pi, p)$;
12: $\lambda \in \Delta^\Pi \leftarrow \arg\min_{\lambda' \in \Delta^\Pi} \max_{\pi^* \in \Pi} \sum_{s,a,h} \frac{d_p^{\pi^*}(s,a,h)}{\sum_\pi \lambda'_\pi d_p^\pi(s,a,h)}$;
13: **return:** $\bar{\pi}$ be the mixed policy which plays $\pi \in \Pi$ with probability $\lambda_\pi$;

---

Based on Lemma 4 and Lemma 5, we summarize the performance of Algorithm 3 as below.

**Lemma 6.** *Let $\Pi_{\text{next}} = Traj\text{-}Learning(p, T, \Pi)$. Fix $\tilde{x}, \tilde{y} \geq 0$. Assume that*

- $\pi^* \in \Pi$;

- *$p$ is an $(3, \tilde{x})$-approximation of $P$ with respect to $\Pi$;*

- *$W^\pi(R, P) \geq W^*(R, P) - \tilde{y}$ for any $\pi \in \Pi$;*

- $\kappa \geq 20 \left( 72\sqrt{\frac{SAH^3\iota}{T}} + 6\sqrt{\frac{S^2AH^2\tilde{y}\iota}{T}} + \frac{100S^2AH^3\iota}{T} + SAH^2\tilde{x}\iota \right).$

*With probability $1 - \delta$, it holds that $\pi^* \in \Pi$ and*

$$W^\pi(R, P) \geq W^*(R, P) - 2\kappa$$

*for any $\pi \in \Pi_{\text{next}}$.*

The full proofs of Lemma 4, Lemma 5 and Lemma 6 are presented in Appendix D.

## 5.3 THE FINAL REGRET BOUND

**Theorem 7.** *Fix $\delta > 0$. For any episodic MDP with trajectory feedback, with probability $1 - \delta$, the regret in $K$ episodes of Algorithm 1 does not exceeds*

$$\text{Regret}(K) \leq \tilde{O}\left( \sqrt{SAH^3K} + \sqrt{S^3A^2H^3}K^{\frac{3}{8}} + \sqrt{S^{11}A^3H^{17}}K^{\frac{1}{4}} + \sqrt{S^{17}A^3H^{27}} \right).$$

By Lemma 3 and the fact that $\pi^* \in \Pi_{\text{det}}$ with probability $1 - \delta$, we have that

- $\pi^* \in \Pi_1$;
- $\tilde{P}$ is an $(3, \sigma_0)$-approximation of $P$ with respect to $\Pi_1$, hence it is also an $(3, \sigma_0)$-approximation of $P$ with respect to $\Pi_\ell$ for any $\ell \geq 1$;
- $W^\pi(R, P) \geq W^*(R, P) - 2\epsilon_0$ for all $\pi \in \Pi_1$.

By the third property, the regret in the first $K_0$ episodes is bounded by

$$\tilde{O}\left(K_0 \epsilon_0 + H\sqrt{SAHK}\right) = \tilde{O}\left(\sqrt{SAH^3K} + S^{\frac{11}{2}}A^{\frac{3}{2}}H^{\frac{17}{2}}K^{\frac{1}{4}} + S^{\frac{17}{2}}A^{\frac{3}{2}}H^{\frac{27}{2}}\right). \quad (14)$$

Now we fix $1 \leq \ell \leq L$ and assume $\pi^* \in \Pi_\ell$. Set $\tilde{x} = \sigma_0$, $\tilde{y} = 2\epsilon_0$, $\iota = \log^2\left(\frac{16SAHT}{\delta}\right)$ and

$$\epsilon_\ell = 20\left(72\sqrt{\frac{SAH^3\iota}{K_\ell}} + 9\sqrt{\frac{S^2AH^2\epsilon_0\iota}{K_\ell}} + \frac{100S^2AH^3\iota}{K_\ell} + SAH^2\sigma_0\iota\right).$$

We then can verify the conditions in Lemma 6: (1) $\pi^* \in \Pi_\ell$; (2) $\tilde{P}$ is an $(3, \tilde{x})$-approximation of $P$ with respect to $\Pi_\ell$; (3) $W^\pi(R, P) \geq W^*(R, P) - \tilde{y}$ for any $\pi \in \Pi_\ell$; (4) $\epsilon_\ell \geq 20\left(72\sqrt{\frac{SAH^3\iota}{K_\ell}} + 6\sqrt{\frac{S^2AH^2\tilde{y}\iota}{K_\ell}} + \frac{100S^2AH^3\iota}{K_\ell} + SAH^2\tilde{x}\iota\right)$.

Using Lemma 6, with probability $1 - \delta$, it holds that: (1) $\pi^* \in \Pi_{\ell+1}$; (2) $W^\pi(R, P) \geq W^*(R, P) - 2\epsilon_\ell$ for any $\pi \in \Pi_{\ell+1}$. By induction on $\ell = 1, 2, \ldots, L$, with probability $1 - \frac{L\delta}{(L+1)}$, it holds that

$$W^\pi(R, P) \geq W^*(R, P) - 2\epsilon_\ell.$$

Recalling that $K_\ell = 2K^{1-\frac{1}{2^l}}$ for $1 \leq \ell \leq L - 1$ and $K_L \leq 2K^{1-\frac{1}{2^L}}$, the regret in the $\ell$-th batch is bounded by

$$O(K_\ell \epsilon_{\ell-1}) = \tilde{O}\left(\sqrt{SAH^3K} + \sqrt{S^3A^2H^3}K^{\frac{3}{8}} + \sqrt{S^6A^2H^7}K^{\frac{1}{4}} + S^2AH^3K^{\frac{1}{2^\ell}}\right)$$

for $2 \leq \ell \leq L$. For $\ell = 1$, the regret in the $\ell$-th batch is bounded by $O(K_1H) = O(\sqrt{KH^2})$. Putting all together, we obtain that the total regret is bounded by

$$\tilde{O}\left(\sqrt{SAH^3K} + \sqrt{S^3A^2H^3}K^{\frac{3}{8}} + \sqrt{S^{11}A^3H^{17}}K^{\frac{1}{4}} + \sqrt{S^{17}A^3H^{27}}\right).$$

The proof is finished by replaced $\delta$ with $\frac{\delta}{16S^2AH(L+1)}$.

## 6 CONCLUSION

In this work, we design an algorithm to achieve asymptotic optimal regret bound of $\tilde{O}(\sqrt{SAH^3K})$ for reinforcement learning with trajectory feedback. However, the proposed algorithm is based on elimination, resulting exponential time cost. It poses a challenge to ascertain whether achieving the optimal regret bound is viable using a more efficient algorithm. Additionally, an interesting direction involves minimizing the lower-order terms in the regret bound.

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

## A   PARAMETER SETTINGS.

Set $K_0 = 100000 S^{\frac{9}{2}} A^{\frac{3}{2}} H^{\frac{17}{2}} K^{\frac{1}{2}} \log\left(\frac{SAHK}{\delta}\right)$ and $K_\ell = 2K^{1-\frac{1}{2^\ell}}$ for $\ell \geq 1$. Let $L := \min_{\ell'}(K_0 + \sum_{\ell=1}^{\ell'} K_\ell) \geq K$. Set $\epsilon_0 = 90000 \log^3\left(\frac{SAH}{\delta}\right)\left(\frac{SAH}{K^{\frac{1}{4}}} + \frac{S^4 AH^6}{K^{\frac{1}{2}}}\right)$, $\sigma_0 = \frac{1}{S^{\frac{3}{2}} A^{\frac{1}{2}} H^{\frac{7}{2}} K^{\frac{1}{2}}}$, $\iota = \log^2\left(\frac{16SAHT}{\delta}\right)$ and $\kappa = 20\left(72\sqrt{\frac{SAH^3\iota}{T}} + 9\sqrt{\frac{S^2 AH^2 \epsilon_0 \iota}{T}} + \frac{100 S^2 AH^3 \iota}{T} + SAH^2 \sigma_0 \iota\right)$.

By this definition, we have $L \leq 2\log_2 \log(K)$. With a slightly abuse of notation, we re-define $K_L = K - (K_0 + \sum_{\ell=1}^{L-1} L_\ell)$. It then holds that $K_0 + \sum_{\ell=1}^{L} K_\ell = K$.

## B   TECHNICAL LEMMAS

**Lemma 8.** *For any policy set $\Pi \subset \Pi_{\text{det}}$,*

$$\min_{\bar\pi \in \Delta^\Pi} \max_{\pi \in \Pi} \sum_{\tau \in \mathcal{T}} \Pr_{\pi,P}[\tau] \phi_\tau^\top (\Lambda(\bar\pi))^{-1} \phi_\tau = SAH, \tag{15}$$

*where $\Lambda(\pi) := \sum_{\tau \in \mathcal{T}} \Pr_{\pi,P}[\tau] \phi_\tau \phi_\tau^\top$.*

*Proof of Lemma 8.* Let $F(\pi) := \log(\det(\Lambda(\pi)))$ for $\pi \in \Delta^\Pi$. Because $\Delta^\Pi$ is a closed set and $F(\pi) \leq d \log(d)$ for any $\pi \in \Delta^\Pi$ with $d = SAH$, there exists some $\bar\pi$ such that $\bar\pi = \arg\max_{\pi \in \Delta^\Pi} F(\pi)$. We assume that $\det(\Lambda(\bar\pi)) \neq 0$. Otherwise $\det(\Lambda(\pi))$ is always 0, which implies there is redundant dimension. Let $\lambda(\bar\pi, \pi)$ be the probability that $\bar\pi$ distributes on $\pi$ for $\pi \in \Pi$. For two different $\pi_1, \pi_2 \in \Pi$ such that $\lambda(\bar\pi, \pi_1) > 0, \lambda(\bar\pi, \pi_2) > 0$, by the condition that $\bar\pi = \arg\max_{\pi \in \Delta^\Pi} F(\pi)$, we have that

$$\frac{\partial F(\bar\pi)}{\partial \lambda(\bar\pi, \pi_1)} = \frac{\partial F(\bar\pi)}{\partial \lambda(\bar\pi, \pi_2)}, \tag{16}$$

which means that

$$\sum_{\tau \in \mathcal{T}} \Pr_{\pi_1,P}[\tau] \phi_\tau^\top (\Lambda(\bar\pi))^{-1} \phi_\tau = \sum_{\tau \in \mathcal{T}} \Pr_{\pi_2,P}[\tau] \phi_\tau^\top (\Lambda(\bar\pi))^{-1} \phi_\tau.$$

For $\pi_1, \pi_2$ such that $\lambda(\bar\pi, \pi_1) > 0$ and $\lambda(\bar\pi, \pi_2) = 0$, we have that

$$\frac{\partial F(\bar\pi)}{\partial \lambda(\bar\pi, \pi_1)} \geq \frac{\partial F(\bar\pi)}{\partial \lambda(\bar\pi, \pi_2)},$$

which implies

$$\sum_{\tau \in \mathcal{T}} \Pr_{\pi_1,P}[\tau] \phi_\tau^\top (\Lambda(\bar\pi))^{-1} \phi_\tau \geq \sum_{\tau \in \mathcal{T}} \Pr_{\pi_2,P}[\tau] \phi_\tau^\top (\Lambda(\bar\pi))^{-1} \phi_\tau.$$

Therefore, $\max_{\pi \in \Pi} \sum_{\tau \in \mathcal{T}} \Pr_{\pi,P}[\tau] \phi_\tau^\top (\Lambda(\bar\pi))^{-1} \phi_\tau$ is reached by any $\pi$ such that $\lambda(\bar\pi, \pi) > 0$. Assume this value is $x$. That is,

$$\lambda(\bar\pi, \pi) \sum_{\tau \in \mathcal{T}} \Pr_{\pi,P}[\tau] \phi_\tau^\top (\Lambda(\bar\pi))^{-1} \phi_\tau = \lambda(\bar\pi, \pi) x$$

for all $\pi \in \Pi$. Taking sum over $\pi \in \Pi$, we have that

$$x = \text{Trace}(\Lambda(\bar\pi)(\Lambda(\bar\pi))^{-1}) = d = SAH. \tag{17}$$

The proof is completed. $\square$

**Lemma 9** (Lemma 1 in Zhang et al. (2022b)). *Let $d > 0$ be an integer. Let $\mathcal{X} \subset (\Delta^d)^m$. Then there exists a distribution $\mathcal{D}$ over $\mathcal{X}$, such that*

$$\max_{x=\{x_i\}_{i=1}^{dm} \in \mathcal{X}} \sum_{i=1}^{dm} \frac{x_i}{y_i} = md,$$

*where $y = \{y_i\}_{i=1}^{dm} = \mathbb{E}_{x \sim \mathcal{D}}[x]$.*

**Lemma 10** (Bennet's inequality). *Let $Z, Z_1, ..., Z_n$ be i.i.d. random variables with values in $[0, 1]$ and let $\delta > 0$. Define $\mathbb{V}Z = \mathbb{E}\left[(Z - \mathbb{E}Z)^2\right]$. Then we have*

$$\mathbb{P}\left[\left|\mathbb{E}\left[Z\right] - \frac{1}{n}\sum_{i=1}^{n} Z_i\right| > \sqrt{\frac{2\mathbb{V}Z\ln(2/\delta)}{n}} + \frac{\ln(2/\delta)}{n}\right] \leq \delta.$$

**Lemma 11** (Theorem 4 in Maurer & Pontil (2009) ). *Let $Z, Z_1, ..., Z_n$ ($n \geq 2$) be i.i.d. random variables with values in $[0, 1]$ and let $\delta > 0$. Define $\bar{Z} = \frac{1}{n}\sum_{i=1}^{n} Z_i$ and $\hat{V}_n = \frac{1}{n}\sum_{i=1}^{n}(Z_i - \bar{Z})^2$. Then we have*

$$\mathbb{P}\left[\left|\mathbb{E}\left[Z\right] - \frac{1}{n}\sum_{i=1}^{n} Z_i\right| > \sqrt{\frac{2\hat{V}_n\ln(2/\delta)}{n-1}} + \frac{7\ln(2/\delta)}{3(n-1)}\right] \leq \delta.$$

**Lemma 12** (Lemma 10 in (Zhang et al., 2022a)). *Let $X_1, X_2, \ldots$ be a sequence of random variables taking value in $[0, l]$. For any $k \geq 1$, let $\mathcal{F}_k$ be the $\sigma$-algebra generated by $(X_1, X_2, \ldots, X_k)$, and define $Y_k := \mathbb{E}[X_k \mid \mathcal{F}_{k-1}]$. Then for any $\delta > 0$, we have*

$$\mathbb{P}\left[\exists n, \sum_{k=1}^{n} X_k \geq 3\sum_{k=1}^{n} Y_k + l\log\frac{1}{\delta}\right] \leq \delta$$

$$\mathbb{P}\left[\exists n, \sum_{k=1}^{n} Y_k \geq 3\sum_{k=1}^{n} X_k + l\log\frac{1}{\delta}\right] \leq \delta.$$

**Lemma 13.** *Fix $d > 0$. Let $\Lambda \in \mathbb{R}^{d \times d}$ be a PSD matrix and $x \in \mathbb{R}^d$ be a vector such that $x^\top \Lambda^{-1} x \leq 1$. Then we have that*

$$\log(\det(\Lambda + xx^\top)) - \log(\det(\Lambda)) \geq 2x^\top \Lambda^{-1} x.$$

*Proof.* Direct computation gives that

$$\log(\det(\Lambda + xx^\top)) - \log(\det(\Lambda)) = \log(\det(\mathbf{I} + x^\top \Lambda^{-1} x^\top)) = \log(1 + x^\top \Lambda^{-1} x) \geq \frac{1}{2}x^\top \Lambda^{-1} x.$$

$\square$

**Lemma 14** (Lemma 20 in Zhang et al. (2021)). *Consider a sequence of independent PSD (positive semi-definite) matrices $X_1, X_2, \ldots, X_n \in \mathbb{R}^{d \times d}$ such that $X_k \preccurlyeq W$ for a fixed PSD matrix $W$ and all $1 \leq k \leq n$. For every $\delta > 0$ and $\epsilon \in (0, 1)$, it holds that*

$$\Pr\left[\sum_{k=1}^{n} X_k \preccurlyeq 3\sum_{k=1}^{n} \mathbb{E}[X_k] + 3\log(d/\delta)W\right] \geq 1 - \delta; \tag{18}$$

$$\Pr\left[\sum_{k=1}^{n} X_k \succcurlyeq \frac{1}{3}\sum_{k=1}^{n} \mathbb{E}[X_k] - 3\log(d/\delta)W\right] \geq 1 - \delta. \tag{19}$$

**Lemma 15.** *Assume $p$ is an $(n, x)$-approximation of $p'$ with respec to $\Pi$. It then holds that*

$$\frac{1}{n}\mathbb{E}_{\pi,p}[\mathbb{I}[(s_h, a_h) = (s, a)]] \leq \mathbb{E}_{\pi,p'}[\mathbb{I}[(s_h, a_h) = (s, a)]] \leq n\mathbb{E}_{\pi,p}[\mathbb{I}[(s_h, a_h) = (s, a)]] + x \tag{20}$$

*for any $\pi \in \Pi$ and $(s, a, h)$.*

*Proof.* By equation 11 and equation 12, for any trajectory $\tau$, we have that $\frac{1}{n}\Pr_p[\tau] \leq \Pr_{p'}[\tau']$. It then holds that

$$\frac{1}{n}\mathbb{E}_{\pi,p}[\mathbb{I}[(s_h, a_h) = (s, a)]] \leq \mathbb{E}_{\pi,p'}[\mathbb{I}[(s_h, a_h) = (s, a)]].$$

On the other hand,

$$\mathbb{E}_{\pi,p'}[\mathbb{I}[(s_h, a_h) = (s, a)]]$$

$$\leq \mathbb{E}_{\pi,p'}[\mathbb{I}[(s_h, a_h) = (s, a)] \cap \mathbb{I}[(s_{h'}, a_{h'}, s_{h+1}, h') \in \mathcal{K}, \forall 1 \leq h' \leq h]] + \max_{\pi \in \Pi_{\det}} \Pr_{\pi,p'}[\mathcal{K}^{\mathrm{C}}]$$

$$\leq n\mathbb{E}_{\pi,p}[\mathbb{I}[(s_h, a_h) = (s, a)]] + x. \tag{21}$$

$\square$

---

**Algorithm 6** Raw-Exploration($\Pi, T, \delta$)

---

1: **Input**: policy set $\Pi$, length $T$, failure probability $\delta$;

2: **Initialize:** $T_1 \leftarrow \frac{T}{SAH}, \iota \leftarrow \log\left(\frac{2S^2AH^2}{\delta}\right), \mathcal{D} \leftarrow \emptyset$;

3: **for** $h = 1, 2, \ldots, H$ **do**

4:     $\mathcal{P} \leftarrow$ Confidence-Region($\mathcal{D}$);

5:     **for** $(s, a) \in \mathcal{S} \times \mathcal{A}$ **do**

6:         $\{\pi^{h,s,a}, p^{h,s,a}\} \arg\max_{\pi \in \Pi, p \in \mathcal{P}} \mathbb{E}_{\pi,p}\left[\mathbb{I}[(s_h, a_h) = (s, a)]\right]$;

7:     **end for**

8:     **for** $(s, a, h) \in \mathcal{S} \times \mathcal{A} \times [H]$ **do**

9:         Execute $\pi^{h,s,a}$ for $T_1$ episodes, and collect the samples as $\mathcal{D}_{h,s,a}$;

10:     **end for**

11:     $\mathcal{D} \leftarrow \mathcal{D} \cup (\cup_{s,a,h} \mathcal{D}_{h,s,a})$;

12: **end for**

13: $\mathcal{P} \leftarrow$ Confidence-Region($\mathcal{D}$);

14: $p \leftarrow$ arbitrary element in $\mathcal{P}$

15: **return**: $p$;

16: **Function**: Confidence-Region($\mathcal{D}$):

17:     $N_h(s, a, s') \leftarrow$ count of $(s, a, h, s')$ in $\mathcal{D}$, for all $(s, a, s')$;

18:     $N_h(s, a) \leftarrow \max\{\sum_{s'} N_h(s, a, s'), 1\}$ for all $(s, a)$;

19:     $\hat{p}_{s,a,h,s'} \leftarrow \frac{N_h(s,a,s')}{N_h(s,a)}, \forall(s, a, h, s')$;

20:     $\mathcal{W} \leftarrow \{(s, a, h, s') : N_h(s, a, s') \geq 200H^2\iota\}$;

21:     $\tilde{\mathcal{P}}_{s,a,h} \leftarrow \left\{p \in \Delta^S | |p_{s'} - \hat{p}_{s,a,h,s'}| \leq \sqrt{\frac{4N_h(s,a,s')\iota}{N_h^2(s,a)}} + \frac{5\iota}{N_h(s,a)}, \forall s' \in \mathcal{S}\right\}, \forall(h, s, a)$;

22:     $\mathcal{P}_{h,s,a} \leftarrow \{\texttt{clip}(p, \mathcal{W}) : p \in \tilde{\mathcal{P}}_{h,s,a}\}, \forall(h, s, a)$;

23:     **Return**: $\otimes_{h,s,a} \mathcal{P}_{s,a,h}$.

24: **Function**: clip($p, \mathcal{W}$)

25:     $p'_{s,a,h,s'} \leftarrow p_{s,a,h,s'}, \forall(h, s, a, s) \in \mathcal{W}$;

26:     $p'_{s,a,h,s'} \leftarrow 0, \forall(s, a, h, s') \notin \mathcal{W}$;

27:     $p'_{s,a,h,z} \leftarrow \sum_{s':(s,a,h,s') \notin \mathcal{W}} p_{s,a,h,s'}, \forall(h, s, a) \in [H] \times \mathcal{S} \times \mathcal{A}$;

28:     $p'_{z,a,h} \leftarrow \mathbf{1}_z, \forall(h, a) \in [H] \times \mathcal{A}$;

29:     **Return**: $p$.

---

**Lemma 16.** *Assume $p$ is an $(n, x)$-approximation of $p'$. It then holds that*

$$\max_{\pi \in \Pi_{\text{det}}} \Pr_{\pi,p'}[\mathcal{T}_{\text{bad}}] \leq x,$$

*where $\mathcal{T}_{\text{bad}} := \{\tau : \Pr_{p'}[\tau] \geq n\Pr_p[\tau]\}$.*

*Proof.* Let $\tau = \{s_h, a_h\}_{h=1}^H$ be an element in $\mathcal{T}_{\text{bad}}$. By definition, there exists $h$ such that $(s_h, a_h, h, s_{h+1}) \in \mathcal{K}^{\text{C}}$. As a result, $\max_{\pi \in \Pi_{\text{det}}} \Pr_{\pi,p'}[\mathcal{T}_{\text{bad}}] \leq \max_{\pi \in \Pi_{\text{det}}} \text{pr}_{\pi,p'}[\mathcal{K}^{\text{C}}] \leq x$. $\square$

## C THE RAW−EXPLORATION ALGORITHM AND ANALYSIS

**Lemma 17.** *By running Raw-Exploration with input $(\Pi, T, \delta)$, with probability $1 - \delta$, the output $p$ is an $\left(3, \frac{11000S^3AH^4\log(SAH/\delta)}{T}\right)$-approximation of $P$ with respect to $\Pi$.*

*Proof.* Let $\mathcal{D}^h$ be the value of $\mathcal{D}$ after the $h$-th iteration. Let $\mathcal{P}^h =$ Confidence-Region($\mathcal{D}^h$) and $\tilde{\mathcal{P}}$ be the final value of $\mathcal{P}$. Let $N_{h'}^h(s, a, s')$ be the count of $(s, a, h', s')$ in $\mathcal{D}_h$ and $N_{h'}^h(s, a) := \min\{\sum_{s'} N_{h'}^h(s, a, s'), 1\}$. Let $\hat{p}_{s,a,h'}^h = \frac{N_{h'}^h(s,a,s')}{N_{h'}^h(s,a)}$ be the empirical transition probability computed by $\mathcal{D}_h$.

By Lemma 10, with probability $1 - \delta/2$,

$$\left|\hat{p}^h_{s,a,h',s'} - P_{s,a,h,s'}\right| \leq \sqrt{\frac{4N^h_{h'}(s,a,s')\iota}{(N^h_{h'}(s,a))^2}} + \frac{5\iota}{N^{h'}_h(s,a)} \tag{22}$$

holds for all $(s,a,h',s')$ and $h \in [H]$. We proceeds the analysis conditioned on equation 22. Let $N_h(s,a,s')$ denote the count of $(s,a,h,s')$ in $\mathcal{D}_{h,s,a}$ and $N_h(s,a) = \max\{\sum_{s'} N_h(s,a,s'), 1\}$. Define

$$\mathcal{K}_h := \{(s,a,s') : N_h(s,a,s') \geq 200H^2\iota\}$$

where $\iota = \log\left(\frac{2S^2AH^2}{\delta}\right)$.

By equation 22, for any $(s,a,s') \in \mathcal{K}_h$ and any $h' \geq h$, we have that

$$\left|\hat{p}^{h'}_{s,a,h,s'} - P_{s,a,h,s'}\right| \leq \hat{p}^{h'}_{s,a,h,s'} \cdot \left(\sqrt{\frac{1}{50H^2}} + \frac{1}{40H^2}\right),$$

which implies that

$$\left|\hat{p}^{h'}_{s,a,h,s'} - P_{s,a,h,s'}\right| \leq \frac{1}{6H}P_{s,a,h,s'}. \tag{23}$$

Moreover, by definition of $\mathcal{P}^h$, using similar arguments, we have

$$\left|p_{s,a,h,s'} - P_{s,a,h,s'}\right| \leq \frac{1}{3H}P_{s,a,h,s'} \tag{24}$$

for any $(s,a,h,s') \in \mathcal{K}_h$ and $p \in \mathcal{P}^h$.

We set $\mathcal{K} = \cup_h \mathcal{K}_h$ and verify the three conditions in Definition 2. The first condition equation 11 holds by equation 24, and the second condition equation 12 holds because $p_{s,a,h,s'} = 0$ for any $p \in \mathcal{P}$ and $(s,a,s') \in \mathcal{K}_h^C$. As for the third condition equation 13, we analyze as below.

Fix $h \in [H]$. By equation 23 and definition of $\{\pi^{h+1,s,a}, p^{h+1,s,a}\}$, we have that

$$\mathbb{E}_{\pi^{h+1,s,a},P}\left[\mathbb{I}[(s_{h+1}, a_{h+1}) = (s,a)]\right]$$

$$\geq \left(1 - \frac{1}{3H}\right)^H \mathbb{E}_{\pi^{h+1,s,a},p^{h+1,s,a}}\left[\mathbb{I}[(s_{h+1}, a_{h+1}) = (s,a)]\right]$$

$$\geq \frac{1}{3}\mathbb{E}_{\pi^{h+1,s,a},p^{h+1,s,a}}\left[\mathbb{I}[(s_{h+1}, a_{h+1}) = (s,a)]\right]$$

$$\geq \frac{1}{3}\max_{\pi \in \Pi}\mathbb{E}_{\pi,p^{h+1,s,a}}\left[\mathbb{I}[(s_{h+1}, a_{h+1}) = (s,a)]\right]$$

$$\geq \frac{1}{9}\max_{\pi \in \Pi}\mathbb{E}_{\pi,P}\left[\mathbb{I}[(s_{h'}, a_{h'}, s_{h'+1}) \in \mathcal{K}_h, \forall 1 \leq h' \leq h] \cdot \mathbb{I}[(s_{h+1}, a_{h+1}) = (s,a)]\right]. \tag{25}$$

Here equation 25 holds because for any trajectory $\tau = \{s_{h'}, a_{h'}\}^h_{h'=1}$ such that $(s_{h'}, a_{h'}, s_{h'+1}) \in \mathcal{K}_{h'}$, $\Pr_{\pi,p}[\tau] \geq \frac{1}{3}\Pr_{\pi,P}[\tau]$ for any $p \in \mathcal{P}^h$ and any $\pi \in \Pi$. Consequently,

$$\mathbb{E}_{\pi^{h+1,s,a},P}\left[\mathbb{I}[(s_{h+1}, a_{h+1}, s_{h+2}) = (s,a,s')]\right]$$

$$\geq \frac{1}{9}\max_{\pi,P}\max_{\pi \in \Pi}\mathbb{E}_{\pi,P}\left[\mathbb{I}[(s_{h'}, a_{h'}, s_{h'+1}) \in \mathcal{K}_h, \forall 1 \leq h' \leq h] \cdot \mathbb{I}[(s_{h+1}, a_{h+1}, s_{h+2}) = (s,a,s')]\right]. \tag{26}$$

On the other side, by Lemma 12, with probability $1 - \frac{\delta}{2S^2AH^2}$, it holds that

$$N_{h+1}(s,a,s')$$

$$\geq \frac{1}{3}T_1\mathbb{E}_{\pi^{h+1,s,a},P}\left[\mathbb{I}[(s_{h+1}, a_{h+1}, s_{h+2}) = (s,a,s')]\right] - \log\left(\frac{2S^2AH^2}{\delta}\right)$$

$$\geq \frac{1}{27}T_1\max_{\pi \in \Pi}\mathbb{E}_{\pi,P}\left[\mathbb{I}[(s_{h'}, a_{h'}, s_{h'+1}) \in \mathcal{K}_h, \forall 1 \leq h' \leq h] \cdot \mathbb{I}[(s_{h+1}, a_{h+1}) = (s,a)]\right] - \log\left(\frac{2S^2AH^2}{\delta}\right),$$

which implies that

$$\max_{\pi \in \Pi} \mathbb{E}_{\pi,P}\left[\mathbb{I}[(s_{h'}, a_{h'}, s_{h'+1}) \in \mathcal{K}_h, \ \forall 1 \le h' \le h] \cdot \mathbb{I}[(s_{h+1}, a_{h+1}) = (s, a)]\right] \le \frac{5427 H^2 \iota}{T_1} \quad (27)$$

for $(s, a, s') \in \mathcal{K}_{h+1}^{\mathrm{C}}$

Taking sum over all $(s, a, s') \in \mathcal{K}_{h+1}^{\mathrm{C}}$, we learn that

$$\max_{\pi \in \Pi} \mathbb{E}_{\pi,P}\left[\mathbb{I}[(s, a, s') \in \mathcal{K}_{h+1}^{C}] \cdot \mathbb{I}[s_{h'}, a_{h'}, s_{h'+1}) \in \mathcal{K}_h, \ \forall 1 \le h' \le h]\right] \le \frac{5427 S^2 A H^2 \iota}{T_1}. \quad (28)$$

Taking sum over $h \in [H]$, we learn that

$$\max_{\pi} \Pr_{\pi,P}[\cup_h \mathcal{K}_h^{\mathrm{C}}]$$
$$\le \sum_{h=1}^{H} \max_{\pi \in \Pi} \mathbb{E}_{\pi,P}\left[\mathbb{I}[(s, a, s') \in \mathcal{K}_{h+1}^{C}] \cdot \mathbb{I}[s_{h'}, a_{h'}, s_{h'+1}) \in \mathcal{K}_h, \ \forall 1 \le h' \le h]\right]$$
$$\le \frac{5427 S^2 A H^3 \iota}{T_1}.$$

Therefore equation 13 holds with $x = \frac{5427 S^2 A H^3 \iota}{T_1}$. The proof is completed by noting $T_1 = \frac{T}{SAH}$.

$\square$

## D  MISSING ALGORITHMS AND PROOFS

### D.1  PROOF OF LEMMA 3

*Proof.* By Lemma 17, with probability $1 - \frac{\delta}{4(L+1)}$, $\tilde{P}$ is an $(3, \frac{11000 S^3 A H^3 \log(4SAH(L+1)/\delta)}{\bar{K}_4})$-approximation of $P$ with respect to $\Pi_1$. By noting that

$$\bar{K}_4 \ge 96000 S^{\frac{9}{2}} A^{\frac{3}{2}} H^{\frac{15}{2}} K^{\frac{1}{2}} \log\left(\frac{SAHK}{\delta}\right)$$

and

$$\sigma_0 \ge \frac{11000 S^3 A H^3 \log(4SAH(L+1)/\delta)}{\bar{K}_4},$$

we conclude that $\tilde{P}$ is an $(3, \sigma_0)$-approximation of $P$ with respect to $\Pi_1$, and thus is an $(3, \sigma_0)$-approximation of $P$ with respect to $\Pi_\ell$ for any $\ell \ge 1$.

Let $b_1 := \frac{11000 S^3 A H^4 \log\left(\frac{4SAH}{\delta}\right)}{\bar{K}_1}$. By Lemma 17, with probability $1 - \frac{\delta}{4}$ $\hat{P}_1$ is an $(3, b_1)$-approximation of $P$ with respect to $\Pi_{\det}$. By Lemma 4, with probability $1 - \frac{\delta}{4}$, we learn that

$$\max_{\pi \in \Pi_{\det}} \left| W^\pi(\hat{R}, P) - W^\pi(R, P) \right|$$
$$\le H \sqrt{\log(SAH) \log(16/\delta)} \left( b_1 + 325 \sqrt{\frac{SAH \log(T) \log(8SAH/\delta)}{\bar{K}_2}} \right)$$
$$\le 1000 \log^2\left(\frac{SAH}{\delta}\right) \cdot \left( \frac{SAH}{K^{\frac{1}{4}}} + 4SAH^2 b_1 \right). \quad (29)$$

By Lemma 5 with parameters as:

$$\Pi = \Pi_{\det};$$

$$x = b_1 = \frac{11000 S^3 A H^4 \log\left(\frac{4SAH}{\delta}\right)}{\bar{K}_1};$$

$$y = H;$$

$$z := 1000 \log^2(\frac{SAH}{\delta}) \cdot \left(\frac{SAH}{K^{\frac{1}{4}}} + \frac{S^4 A H^6}{K^{\frac{1}{2}}}\right)$$

$$b = 30\sqrt{\frac{2S^2 A H^2 \log\left(\frac{32SAH}{\delta}\right)}{\bar{K}_3} + \frac{360 S^2 A H^3 \log\left(\frac{32SAH}{\delta}\right)}{\bar{K}_3} + \frac{44000 S^3 A H^4 \log\left(\frac{32 S^2 A H^2}{\delta}\right)}{\bar{K}_1}};$$

$$\epsilon = \epsilon_0 = 90000 \log^3(\frac{SAH}{\delta}) \left(\frac{SAH}{K^{\frac{1}{4}}} + \frac{S^4 A H^6}{K^{\frac{1}{2}}}\right) \geq 2(b+z) \tag{30}$$

we have that: with probability $1 - \frac{\delta}{4}$, it holds that (1) $\pi^* \in \Pi_1$; (2) $W^\pi(R, P) \geq W^*(R, P) - 2\epsilon$ for any $\pi \in \Pi_1$.

The proof is finished.

$\square$

## D.2 Proof of Lemma 4

*Proof.* Let $d = SAH$. Fix $\pi \in \Pi$. By definition, we have that

$$\left|W^\pi(\hat{R}, P) - W^\pi(R, P)\right| \leq \sum_{\tau \in \mathcal{T}} \Pr_{\pi, P}[\tau] \cdot |\phi_\tau^\top (\hat{R} - R)|.$$

By Lemma 19, with probability $1 - \delta/2$, it holds that $R \in \mathcal{R}$, which implies that

$$\left|W^\pi(\hat{R}, P) - W^\pi(R, P)\right| \leq \sum_{\tau \in \mathcal{T}} \Pr_{\pi, P}[\tau] \cdot |\phi_\tau^\top (\hat{R} - R)|$$

$$\leq \sum_{\tau \in \mathcal{T}} \Pr_{\pi, P}[\tau] \cdot \min\{8\sqrt{H^2 \log(SAH) \log(4/\delta) \phi_\tau^\top \hat{\Lambda}^{-1} \phi_\tau}, H\}$$

$$\leq H\sqrt{\log(SAH) \log(4/\delta)} \sum_{\tau \in \mathcal{T}} \Pr_{\pi, P}[\tau] \min\left\{8\sqrt{\phi_\tau^\top \hat{\Lambda}^{-1} \phi_\tau}, 1\right\} \tag{31}$$

By Lemma 20, with probability $1 - \delta/2$, $\hat{\Lambda} \succcurlyeq 3\tilde{\Lambda}$. Consequently, we have that

$$\left|W^\pi(\hat{R}, P) - W^\pi(R, P)\right| \leq H\sqrt{\log(SAH) \log(4/\delta)} \sum_{\tau \in \mathcal{T}} \Pr_{\pi, P}[\tau] \min\left\{5\sqrt{\phi_\tau^\top \tilde{\Lambda}^{-1} \phi_\tau}, 1\right\}$$

$$\leq H\sqrt{\log(SAH) \log(4/\delta)} \cdot \left(x + 3\Pr_{\pi, p}[\tau] \min\left\{5\sqrt{\phi_\tau^\top \tilde{\Lambda}^{-1} \phi_\tau}, 1\right\}\right) \tag{32}$$

$$\leq H\sqrt{\log(SAH) \log(4/\delta)} \cdot \left(x + 15\sqrt{\Pr_{\pi, p}[\tau] \min\left\{\phi_\tau^\top \tilde{\Lambda}^{-1} \phi_\tau, 1\right\}}\right) \tag{33}$$

$$\leq H\sqrt{\log(SAH) \log(4/\delta)} \cdot \left(x + 325\sqrt{\frac{SAH \log(T) \log(2d/\delta)}{T}}\right). \tag{34}$$

Here equation 32 holds by Lemma 16, equation 33 is by Cauchy's inequality, and equation 34 is by Lemma 18.

The proof is finished.

□

**Lemma 18.** *Let $\tilde{\Lambda}$ be the final value of $\Lambda$ in Algorithm 4. It then holds that*

$$\max_{\pi \in \Pi} \sum_{\tau \in \mathcal{T}} \Pr_{\pi,p}[\tau] \min\{\phi_\tau^\top \tilde{\Lambda}^{-1} \phi_\tau, 1\} \leq \frac{432SAH \log(T) \log(2d/\delta)}{T} \tag{35}$$

*Proof.* Let $T_1 = \frac{T}{54 \log(2d/\delta)}$. Let $\Lambda^t$ be the value of $\Lambda$ before the $t$-th iteration. For any policy $\pi \in \Pi$, we have that

$$\sum_{\tau \in \mathcal{T}} \Pr_{\pi,p}[\tau] \cdot \min\{\phi_\tau^\top \tilde{\Lambda}^{-1} \phi_\tau, 1\} \leq \frac{1}{T_1} \sum_{t=1}^{T_1} \sum_{\tau \in \mathcal{T}} \Pr_{\pi,p}[\tau] \cdot \min\{\phi_\tau^\top (\Lambda^t)^{-1} \phi_\tau, 1\}$$

$$\leq \frac{1}{T_1} \sum_{t=1}^{T_1} \sum_{\tau \in \mathcal{T}} \Pr_{\pi^t,p}[\tau] \cdot \min\{\phi_\tau^\top (\Lambda^t)^{-1} \phi_\tau, 1\}$$

$$\leq \frac{1}{T_1} \cdot 4 \log\left(\frac{\det(\tilde{\Lambda})}{\lambda^{SAH}}\right) \tag{36}$$

$$\leq \frac{432SAH \log(T) \log(2d/\delta)}{T}.$$

Here equation 36 is derived as following. Let $z_{t,\tau} = \phi_\tau \cdot \frac{1}{\sqrt{\max\{\phi_\tau^\top (\Lambda^t)^{-1} \phi_\tau, 1\}}}$. Then we have that $\Lambda^{t+1} = \Lambda^t + \sum_{\tau \in \mathcal{T}} \Pr_{\pi^t,p}[\tau] z_{t,\tau} z_{t,\tau}^\top$. Because $z_{t,\tau} z_{t,\tau}^\top \preccurlyeq \Lambda^t$, it holds that $\sum_{\tau \in \mathcal{T}} \Pr_{\pi^t,p}[\tau] z_{t,\tau} z_{t,\tau}^\top \preccurlyeq \Lambda^t$. Let $\prec$ be an order over all possible trajectories and $\Lambda(\tau) = \Lambda^t + \sum_{\tau' \prec \tau} \Pr_{\pi^t,p}[\tau'] z_{t,\tau'} z_{t,\tau'}^\top) \preccurlyeq 2\Lambda^t$.

As a result, we have that

$$\log\left(\frac{\det(\Lambda^{t+1})}{\det(\Lambda^t)}\right)$$

$$= \sum_{\tau \in \mathcal{T}} \left(\log(\det(\Lambda(\tau) + \Pr_{\pi^t,p}[\tau] z_{t,\tau} z_{t,\tau}^\top) - \log(\det(\Lambda(\tau)))\right)$$

$$\geq \frac{1}{2} \sum_{\tau \in \mathcal{T}} \Pr_{\pi^t,p}[\tau] z_{t,\tau}^\top (\Lambda(\tau))^{-1} z_{t,\tau} \tag{37}$$

$$\geq \frac{1}{4} \sum_{\tau \in \mathcal{T}} \Pr_{\pi^t,p}[\tau] z_{t,\tau}^\top (\Lambda^t)^{-1} z_{t,\tau}. \tag{38}$$

Here equation 37 is by Lemma 13.

□

**Lemma 19.** *With probability $1 - \delta/2$, $R \in \mathcal{R}$.*

*Proof.* Let $\lambda' = 18\lambda \log(2d/\delta)$. It is easy to see $R \in [0,1]^{SAH}$. It suffices to verify that

$$|\phi_\tau^\top R - \phi_\tau^\top \bar{R}| \leq 8\sqrt{H^2 \log(SAH) \log(2/\delta) \phi_\tau^\top \hat{\Lambda}^{-1} \phi_\tau}, \quad \forall \tau.$$

Let $\tau^t = \{(s_h^t, a_h^t)\}_{h=1}^H$. Let $\zeta^t := Y_t - \sum_{h=1}^T R_h(s_h^t, a_h^t)$. Noting that $Y^t = \sum_{h=1}^H r_h(s_h^t, a_h^t)$ where each $r_h(s_h^t, a_h^t)$ are drawn according to $R_h(s_h^t, a_h^t)$ independently, we have

that $\mathrm{E}[\exp(z\zeta^t)] \leq \exp(Hz^2/2)$ for any $z \geq 0$. For fixed $\tau$, we note that

$$\left|\phi_\tau^\top \bar{R} - \phi_\tau^\top R\right| = \left|\phi_\tau^\top \hat{\Lambda}^{-1} \sum_{t=1}^T \phi_{\tau^t}\zeta^t - \lambda'\phi_\tau^\top \hat{\Lambda}^{-1}R\right|$$

$$\leq \left|\phi_\tau^\top \Lambda^{-1} \sum_{t=1}^T \phi_{\tau^t}\zeta^t\right| + \lambda'H\|\phi_\tau \hat{\Lambda}^{-1}\|_2$$

$$\leq \left|\phi_\tau^\top \Lambda^{-1} \sum_{t=1}^T \phi_{\tau^t}\zeta^t\right| + H\sqrt{\lambda'\phi_\tau^\top \hat{\Lambda}^{-1}\phi_\tau} \qquad (39)$$

$$\leq 2\left|\phi_\tau^\top \Lambda^{-1} \sum_{t=1}^T \phi_{\tau^t}\zeta^t\right|. \qquad (40)$$

Here equation 39 holds by the fact that $\hat{\Lambda} - \lambda'\mathbf{I}$ is PSD and equation 40 is by the fact that $18\lambda\log(2d/\delta)H^2 \leq 1$.

Note that $\{\zeta^t\}_{t=1}^T$ does not change the distribution of $\{\phi_{\tau^t}\}_{t=1}^T$. Therefore, it holds that

$$\Pr\left[\left|\phi_\tau^\top \Lambda^{-1} \sum_{t=1}^T \phi_{\tau^t}\zeta^t\right| \geq x \cdot \sqrt{\phi_\tau^\top \hat{\Lambda}^{-1}\phi_\tau}\right] \leq 2\exp\left(-\frac{x^2}{2H}\right). \qquad (41)$$

With a union bound of all possible choices of $\tau$, we learn that, with probability $1 - \delta$, for any $\tau$, it holds that

$$\left|\phi_\tau^\top \bar{R} - \phi_\tau^\top R\right| \leq 8\sqrt{H^2\log(SAH)\log(4/\delta)\phi_\tau^\top \hat{\Lambda}^{-1}\phi_\tau}.$$

The proof is completed. $\qquad\square$

**Lemma 20.** *With probability $1 - \delta/2$, it holds that*

$$\hat{\Lambda} \succcurlyeq 3\tilde{\Lambda}.$$

*Proof.* Let $\Lambda^t \preccurlyeq \tilde{\Lambda}$ be the value of $\Lambda$ before the $t$-th round in line 5. Let $z_t = \phi_{\tau^t}\sqrt{\frac{1}{\max\{\phi_{\tau^t}^\top\tilde{\Lambda}^{-1}\phi_{\tau^t},1\}}}$. It is then easy to verify that $\tilde{\Lambda} \succcurlyeq z_tz_t^\top$. By Lemma 16, we have $\Pr_p[\tau] \leq 3\Pr_{p'}[\tau]$ for any $\tau$. By noting that

$$\tilde{\Lambda} = \sum_{t=1}^{T_1} \mathbb{E}_{\pi^t,p}\left[\sum_{\tau\in\mathcal{T}} \Pr_{\pi^t,p}\phi_\tau\phi_\tau^\top \cdot \frac{1}{\max\{\phi_\tau^\top(\Lambda^t)^{-1}\phi_\tau,1\}}\right]$$

$$\preccurlyeq \sum_{t=1}^{T_1} \mathbb{E}_{\pi^t,p}\left[\sum_{\tau\in\mathcal{T}} \Pr_{\pi^t,p}\phi_\tau\phi_\tau^\top \cdot \frac{1}{\max\{\phi_\tau^\top\tilde{\Lambda}^{-1}\phi_\tau,1\}}\right]$$

$$= T_1\mathbb{E}_{\bar{\pi},p}\left[\sum_{\tau\in\mathcal{T}} \Pr_{\bar{\pi},p}\phi_\tau\phi_\tau^\top \cdot \frac{1}{\max\{\phi_\tau^\top\tilde{\Lambda}^{-1}\phi_\tau,1\}}\right],$$

we have

$$18\log(2d/\delta)\lambda\mathbf{I} + \mathbb{E}_{\pi^t,P}\left[\sum_{t=1}^T z_tz_t^\top\right]$$

$$\succcurlyeq 18\log(2d/\delta)\lambda\mathbf{I} + \frac{1}{3}\mathbb{E}_{\bar{\pi},p}\left[\sum_{t=1}^T \phi_{\tau^t}\phi_{\tau^t}^\top \cdot \frac{1}{\max\{\phi_{\tau^t}^\top\tilde{\Lambda}^{-1}\phi_{\tau^t},1\}}\right]$$

$$\succcurlyeq 18\log(2d/\delta)\tilde{\Lambda}. \qquad (42)$$

By Lemma 14, with probability $1 - \delta/2$,

$$\sum_{t=1}^T z_tz_t^\top \succcurlyeq \frac{1}{3}\mathbb{E}\left[\sum_{t=1}^T z_tz_t^\top\right] - 3\log(2d/\delta)\tilde{\Lambda} \succcurlyeq 3\log(2d/\delta)\tilde{\Lambda} - 18\lambda\log(2d/\delta)\mathbf{I},$$

which means that

$$\hat{\Lambda} \succcurlyeq 18\lambda\log(2d/\delta)\mathbf{I}\sum_{t=1}^{T}z_t z_t^{\top} \succcurlyeq 3\tilde{\Lambda}.$$

The proof is completed. □

### D.3 PROOF OF LEMMA 6

Let $\hat{R}$ be the reward function in line 2 Algorithm 3. By Lemma 4, with probability $1 - \delta/2$,

$$\max_{\pi\in\Pi}\left|W^{\pi}(\hat{R}, P) - W^{\pi}(R, P)\right| \leq b_1 := H\sqrt{\log(SAH)\log(8/\delta)} \cdot \left(\tilde{x} + 325\sqrt{\frac{SAH\log(T)\log\left(\frac{4SAH}{\delta}\right)}{T}}\right).$$

As a result, for any $\pi \in \Pi$,

$$W^{\pi}(\hat{R}, P) - W^{*}(\hat{R}, P) \geq W^{\pi}(R, P) - W^{*}(R, P) - 2\max_{\pi\in\Pi}\left|W^{\pi}(\hat{R}, P) - W^{\pi}(R, P)\right| \geq \tilde{y} + 2b_1.$$
(43)

Let $x = x_1$, $y = \tilde{y} + 2b_1$, $z = b_1$. Let $b = 30\sqrt{\frac{SAH^2(H+Sy)\log\left(\frac{16SAH}{\delta}\right)}{T}} + \frac{360S^2AH^3\log\left(\frac{16SAH}{\delta}\right)}{T} + 4SAH^2\tilde{x}$.

By Lemma 5 and the assumption that $\kappa \geq 2(b + z) = 2(b + b_1)$, it then holds that $\pi^* \in \Pi_{\text{next}}$ and
$$W^{\pi}(R, P) \geq W^{*}(R, P) - 2\kappa$$

for any $\pi \in \Pi_{\text{next}}$

### D.4 PROOF OF LEMMA 5

*Proof of Lemma 5.* In this proof, we use $\{v_h^{\pi}(s)\}$ ($\{v_h^{*}(s)\}$) to denote the (optimal) value function under the policy $\pi$, transition $P$ and reward $u$. With a slight abuse of notation, we define $d_P^{\bar{\pi}}(s, a, h) = \mathbb{E}_{\bar{\pi}, P}\left[\mathbb{I}[(s_h, a_h) = (s, a)]\right]$.

Because $p$ is an $(3, x)$−approximation of $P$ with respect to $\Pi$, by Lemma 15 we have that

$$\frac{1}{3}c(s, a, h) \leq d_P^{\bar{\pi}}(s, a, h) \leq 3c(s, a, h) + x.$$
(44)

Let $\mathcal{L} := \{(s, a, h) : c(s, a, h) \geq \max\{x, \frac{36\log(8SAH/\delta)}{T}\}\}$. By equation 44, $d_P^{\bar{\pi}}(s, a, h) \leq 4x$ for $(s, a, h) \notin \mathcal{L}$. By noting that $\hat{p}_{s_h, a_h, h}$ is independent of $v_{h+1}^{*}$, using Bernstein's inequality, with probability $1 - \delta/8$,

$$\left|(\hat{p}_{s,a,h} - P_{s,a,h})v_{h+1}^{*}\right| \leq 2\sqrt{\frac{\mathbb{V}(P_{s,a,h}, v_{h+1}^{*})\log(8SAH/\delta)}{N_h(s, a)}} + \frac{H\log(8SAh/\delta)}{N_h(s, a)}, \quad \forall(s, a, h);$$
(45)

$$\left|\hat{p}_{s,a,h,s'} - P_{s,a,h,s'}\right| \leq 2\sqrt{\frac{P_{s,a,h,s'}\log(8SAH/\delta)}{N_h(s, a)}} + \frac{H\log(8SAH/\delta)}{N_h(s, a)}, \quad \forall(s, a, h, s').$$
(46)

We continue the analysis conditioned on equation 45 and equation 46. Fix $\pi \in \Pi$. Using policy difference lemma, and noting that $d_P^{\pi}(s, a, h) \leq 4x$ for $(s, a, h) \notin \mathcal{L}$, we have that

$$\left|W^{\pi}(\hat{R}, \hat{p}) - W^{\pi}(\hat{R}, P)\right| = \left|\mathbb{E}_{\pi, P}\left[\sum_{h=1}^{H}(\hat{p}_{s_h, a_h, h} - P_{s_h, a_h, h})v_{h+1}^{\pi}\right]\right|$$

$$\leq \left|\sum_{(s,a,h)\in\mathcal{L}}d_P^{\bar{\pi}}(s, a, h)(\hat{p}_{s,a,h} - P_{s,a,h})v_{h+1}^{\pi}\right| + 4SAH^2\left(x + \frac{36\log(8SAH/\delta)}{T}\right).$$
(47)

Let $F = 4SAH^2 \left( x + \frac{36 \log(8SAH/\delta)}{T} \right)$. By definition of $\mathcal{E}_1$, we further have that,

$$\left| W^\pi(\hat{R}, \hat{p}) - W^\pi(\hat{R}, P) \right| \tag{48}$$

$$\leq \left| \sum_{(s,a,h) \in \mathcal{L}} d_P^\pi(s,a,h)(\hat{p}_{s,a,h} - P_{s,a,h}) v_{h+1}^* \right| + \left| \sum_{(s,a,h) \in \mathcal{L}} d_P^\pi(s,a,h)(\hat{p}_{s,a,h} - P_{s,a,h})(v_{h+1}^\pi - v_{h+1}^*) \right| + F$$

$$\leq \left| \sum_{(s,a,h) \in \mathcal{L}} d_P^\pi(s,a,h) \left( 2\sqrt{\frac{\mathbb{V}(P_{s,a,h}, v_{h+1}^*) \log\left(\frac{8SAH}{\delta}\right)}{N_h(s,a)}} + \frac{H \log\left(\frac{8SAH}{\delta}\right)}{N_h(s,a)} \right) \right|$$

$$+ \left| \sum_{(s,a,h) \in \mathcal{L}} d_P^\pi(s,a,h) \left( 2\sqrt{\frac{S\mathbb{V}(P_{s,a,h}, v_{h+1}^* - v_{h+1}^\pi) \log\left(\frac{8SAH}{\delta}\right)}{N_h(s,a)}} + \frac{SH \log\left(\frac{8SAH}{\delta}\right)}{N_h(s,a)} \right) \right| + F$$

$$\leq 2\sqrt{\log\left(\frac{8SAH}{\delta}\right) \left( \sum_{(s,a,h) \in \mathcal{L}} \frac{d_P^\pi(s,a,h)}{N_h(s,a)} \right) \cdot \sqrt{\mathbb{E}\left[ \sum_{h=1}^{H} \left( \mathbb{V}(P_{s_h,a_h,h}, v_{h+1}^*) + S\mathbb{V}(P_{s_h,a_h,h}, v_{h+1}^* - v_{h+1}^\pi) \right) \right]}}$$

$$+ 2SH \log\left(\frac{8SAH}{\delta}\right) \left( \sum_{(s,a,h) \in \mathcal{L}} \frac{d_P^\pi(s,a,h)}{N_h(s,a)} \right) + F. \tag{49}$$

We then bound the terms in equation 49 separately.

**The doubling count term.** By definition of $\mathcal{L}$, we have that

$$\sum_{(s,a,h) \in \mathcal{L}} \frac{d_P^\pi(s,a,h)}{N_h(s,a)} \leq 4 \sum_{s,a,h} \frac{d_p^\pi(s,a,h)}{N_h(s,a)}. \tag{50}$$

By Lemma 12, we further have that, with probability $1 - \frac{\delta}{8}$, it holds that

$$N_h(s,a) \geq \frac{1}{9} Tc(s,a,h) - \log(8SAH/\delta). \tag{51}$$

for any $(s,a,h)$. Conditioned on this event, we have that

$$\sum_{(s,a,h) \in \mathcal{L}} \frac{d_P^\pi(s,a,h)}{N_h(s,a)} \leq \frac{108}{T} \sum_{s,a,h} \frac{d_p^\pi(s,a,h)}{c(s,a,h)} \leq \frac{108SAH}{T}. \tag{52}$$

In the last inequality, we use the fact that

$$\max_{\pi^* \in \Pi} \sum_{s,a,h} \frac{d_p^{\pi^*}(s,a,h)}{c(s,a,h)} = SAH, \tag{53}$$

which is a direct result following Lemma 9.

**The variance terms.** Direct computation gives that

$$
\mathbb{E}_{\pi,P}\left[\sum_{h=1}^{H}\mathbb{V}(P_{s_h,a_h,h},v_{h+1}^*)\right] = \mathbb{E}_{\pi,P}\left[\sum_{h=1}^{H}\left((v_{h+1}^*(s_{h+1}))^2 - (P_{s_h,a_h,h}v_{h+1}^*)^2\right)\right]
$$

$$
\leq \mathbb{E}_{\pi,P}\left[\sum_{h=1}^{H}\left((v_h^*(s_h))^2 - (P_{s_h,a_h,h}v_{h+1}^*)^2\right)\right]
$$

$$
\leq 2H\mathbb{E}_{\pi,P}\left[\sum_{h=1}^{H}\left(v_h^*(s_h) - P_{s_h,a_h,h}v_{h+1}^*\right)\right]
$$

$$
= 2H\mathbb{E}_{\pi,P}\left[\sum_{h=1}^{H}\left(v_h^*(s_h) - v_{h+1}^*(s_{h+1})\right)\right]
$$

$$
\leq 2H^2 \tag{54}
$$

and

$$
\mathbb{E}_{\pi,\mathbb{P}}\left[\sum_{h=1}^{H}\mathbb{V}(P_{s_h,a_h,h},v_{h+1}^*) + S\mathbb{V}(P_{s_h,a_h,h},v_{h+1}^* - v_{h+1}^\pi)\right]
$$

$$
= \mathbb{E}_{\pi,P}\left[\sum_{h=1}^{H}\left((v_{h+1}^*(s_{h+1}) - v_{h+1}^\pi(s_{h+1}))^2 - (P_{s_h,a_h,h}(v_{h+1}^* - v_{h+1}^\pi))^2\right)\right]
$$

$$
\leq \mathbb{E}_{\pi,P}\left[\sum_{h=1}^{H}\left((v_h^*(s_h) - v_h^\pi(s_h))^2 - (P_{s_h,a_h,h}(v_{h+1}^* - v_{h+1}^\pi))^2\right)\right]
$$

$$
\leq H\mathbb{E}_{\pi,P}\left[\sum_{h=1}^{H}\left|(v_h^*(s_h) - P_{s_h,a_h,h}v_{h+1}^*) - (v_h^\pi(s_h) - P_{s_h,a_h,h}v_{h+1}^\pi)\right|\right]
$$

$$
= 2H\mathbb{E}_{\pi,P}\left[\sum_{h=1}^{H}\left|(v_h^*(s_h) - P_{s_h,a_h,h}v_{h+1}^*) - \hat{R}_h(s_h,a_h)\right|\right]
$$

$$
= 2H\mathbb{E}_{\pi,P}\left[\sum_{h=1}^{H}\left(v_h^*(s_h) - u_h(s_h,a_h) - P_{s_h,a_h,h}v_{h+1}^*\right)\right] \tag{55}
$$

$$
2H(W^*(u,P) - W^\pi(u,P))
$$

$$
\leq 2Hy. \tag{56}
$$

Here equation 55 holds by the fact that $v_h^*(s_h) \geq u_h(s_h,a_h) + P_{s_h,a_h,h}v_{h+1}^*$.

**Putting together.** By equation 49, equation 52 equation 54 and equation 56, we have that

$$
|W^\pi(u,\hat{p}) - W^\pi(u,P)| \leq 30\sqrt{\frac{SAH^2(H+Sy)\log\left(\frac{8SAH}{\delta}\right)}{T}} + \frac{360S^2AH^3\log\left(\frac{8SAH}{\delta}\right)}{T} + 4SAH^2x = b. \tag{57}
$$

Now we verify that $\pi^* \in \Pi_{\text{next}}$.

It suffices to show that

$$
W^{\pi^*}(u,\hat{p}) \geq \max_{\pi' \in \Pi} W^{\pi'}(u,\hat{p}) - \epsilon. \tag{58}
$$

By the assumptions and equation 57, we have that

$$
W^{\pi^*}(u,\hat{p}) \geq W^{\pi^*}(u,P) - b \geq W^{\pi^*}(R,P) - b - z
$$

$$
W^\pi(\mu,\hat{p}) \leq W^\pi(u,P) + b \leq W^\pi(R,P) + b + z \leq W^{\pi^*}(R,P) + b + z.
$$

Noting that $\epsilon \geq 2(b+z)$, we conclude that $\pi^* \in \Pi_{\text{next}}$. On the other hand, for any $\pi \in \Pi_{\text{next}}$, we have that

$$W^{\pi}(R, P) \geq W^{\pi}(u, \hat{p}) - (b+z) \geq W^{\pi^*}(u, \hat{p}) - 2(b+z) \geq W^{\pi^*}(R, P) - 3(b+z) \geq W^{\pi^*}(R, P) - 2\epsilon.$$

The proof is finished. □

