# OpenReview forum: "Minimax Optimal Regret Bound for Reinforcement Learning with Trajectory Feedback"
_ICLR.cc/2025/Conference — Submitted to ICLR 2025_

### Official Review · Reviewer_CXMx · 2024-10-30

**Soundness:** 3
**Presentation:** 3
**Contribution:** 3
**Rating:** 6
**Confidence:** 3

**Summary:**

This paper studies reinforcement learning (RL) with trajectory feedback. The learner can only observe the accumulative noisy reward once a trajectory is terminated. The state transition model is unknown to the learner. The paper develops an online learning algorithm and shows that its regret bound is optimal.

**Strengths:**

1. The paper is well-written and easy to follow.
2. RL with trajectory feedback is an interesting problem.
3. This is a solid theory paper. The analysis is rigorous and sound.

**Weaknesses:**

1. There is no experiment.
2. In the upper bound of Theorem 7, the last three terms dominate. In contrast, the abstract and the introduction claim that the upper bound is determined by the first term. The authors should clarify under what conditions, if any, the first term dominates. If the first term is not asymptotically dominant, the authors should explain why they only focus on the first term in the abstract and the introduction.
3. The introduction claims that the developed algorithm for RL with trajectory feedback achieves the same asymptotically optimal regret bound as the standard RL. The authors should explain why trajectory feedback does not lead to a worse regret bound and what properties of their algorithm allow them to overcome the information disadvantage of only receiving trajectory feedback.
4. Section 3 should clarify that how the expected trajectory reward is a linear function of the state-action visitation frequencies.
5. Some mathematical derivations are not intuitive. The authors can add explanations about what the mathematical properties mean and how they are derived. Here are some examples.
5.1 The second key observation on page 5.
5.2 Inequality (3).
5.3 The equation in (8).
6. There are a few typos: The third term of the upper bound in Theorem 7. P1 in Line 4 of Algorithm 2. D2 in Line 6 of Algorithm 2.

**Questions:**

1. In the upper bound of Theorem 7, the last three terms dominate. In contrast, the abstract and the introduction claim that the upper bound is determined by the first term. The authors should clarify under what conditions, if any, the first term dominates. If the first term is not asymptotically dominant, the authors should explain why they only focus on the first term in the abstract and the introduction.
2. The introduction claims that the developed algorithm for RL with trajectory feedback achieves the same asymptotically optimal regret bound as the standard RL. The authors should explain why trajectory feedback does not lead to a worse regret bound and what properties of their algorithm allow them to overcome the information disadvantage of only receiving trajectory feedback.

---

> ### Author Response · Authors · 2024-11-22
> **Response to Reviewer CXMx**
>
> For your questions about (1) the lower order term and (2) the intuition why trajectory-feedback does not hurt the regret performance, please refer to the common response.
>
> **About linearity of trajectory reward in Section 3:** Thanks for the suggestion. We have provided a proof in the revision, which we also present here. $\mathbb{E}[r_{\tau}] = \mathbb{E}[\sum_{h=1}^H r_h] = \sum_{s,a,h}\mathbb{E}[  \mathbb{I}[(s_h,a,_h)=(s,a) ] ]r_h(s,a) = d^{\top}r$.
>
>
>
> **About mathematical derivations in Section 5:** Thanks for the suggestion. We explain as follows. *
>
> **About the second observation in page 5:**
>  $d_{P}^{\pi}(s,a,h):=Pr_{\pi,P}[(s_h,a_h)=(s,a)] = \sum_{\tau}Pr_{\pi,P}[,\tau] \cdot I[ (s_h(\tau),a_h(\tau))=(s,a) ]$, where $(s_{h}(\tau),a_h(\tau))$ is the $h$-th state-action pair in $\tau$.
>
> **About (3):**  This equation is based on confidence regions for linear bandits.
>
> **About (8):** the equation is by definition that $W^{\pi}(r,p) =E_{\pi,p}[\sum_{h=1}^H r_h] = \sum_{s,a,h}d_{p}^{\pi}(s,a,h)\cdot r_h(s,a) = (d_{p}^{\pi})^{\top}r$.

---

> > ### Comment · Reviewer_CXMx · 2024-11-26
> >
> > Thanks for the response. I would like to maintain my rating 6 as the paper does not conduct experiments to show the performance of the developed algorithm.

---

### Official Review · Reviewer_qWPS · 2024-10-31

**Soundness:** 2
**Presentation:** 2
**Contribution:** 3
**Rating:** 3
**Confidence:** 3

**Summary:**

This paper proposes an algorithm that achieves a minimax optimal regret bound for RL with trajectory feedback. Specifically, the proposed algorithm achieves a regret bound of $\tilde{O}(\sqrt{SAH^3K})$ for an episodic MDP.

**Strengths:**

- The paper tackles a well-motivated problem, proposing an asymptotically optimal regret bound algorithm for scenarios with trajectory-level feedback.

- Up to Section 4, the authors clearly present their motivations, objectives and proof sketch, making it easier for readers to grasp the core concepts of the paper.

**Weaknesses:**

The paper has numerous typos and complex, undefined notations, giving the impression of a hastily composed and unpolished draft. The structure and expressions appear to be inspired by [1], yet there are instances of notations that are either not used or are undefined in the context of this paper. A thorough revision to polish the paper would improve its clarity.

In particular, Section 5 lacks logical flow, as many concepts are listed without a thorough theoretical explanation, making it difficult for readers to follow, especially given the insufficient description of each pseudocode. Considering page limitations, substituting theoretical proofs with a more accessible explanation of the pseudocode in the main paper could enhance readability.

The proposed algorithm also appears to have impractical elements. See Questions.

Typos

- Line 46: "ofte" should be "often"

- Algorithm 2: $\mathcal{D}_2$ is not defined

- Algorithm 3, Line 3: The font of “plan” needs correction

- Line 388: "we" should be capitalized to "We"

- Line 483: $S^1 1$ should be corrected to $S^{11}$

- Algorithm 5: $\lambda_\pi$ is not defined.

[1] Zhang, Zihan, et al. "Near-optimal regret bounds for multi-batch reinforcement learning." Advances in Neural Information Processing Systems 35 (2022): 24586-24596.

**Questions:**

- On line 310, optimization is mentioned for the double-state reward function $r(s,s',h,h')$. Could the authors provide more detail on this?

- In Algorithm 4, Line 7, does $\bar{\pi}$ refer to what is stated in Eq (6)? Is it possible to obtain $\bar{\pi}$ in a tractable manner?

---

> ### Author Response · Authors · 2024-11-22
> **Response to Reivewer qWPS**
>
> **About writings:**
> Thanks for the careful review. We have fixed the typos accordingly in the revision. In Algorithm 2, $\mathcal{D}_2$ is redundant  so we remove $\mathcal{D}_2$.
>
> **About $\lambda_{\pi}$ in Algorithm 5:**
> In Algorithm 5, we use $\lambda_{\pi}$ to denote the probability of  $\pi$ following the probability vector $\lambda$ in Line 12.
>
> **About structure of Section 5:**
>  We have presented the high-level idea in Section 4. We borrow the ideas to construct a reference model from [1], and then conduct exploration with the help of the reference model.  We also present the usage of each algorithm and the corresponding lemmas in Section 5.1-2.
>
>
>  **About the double-state reward in Line 310:** In Line 4 Algorithm 4, the optimization problem is $\max_{\pi}\Pr_{\pi}[\tau]   \min\\{   \phi^{\top}\Lambda^{-1}\phi   , 1\\}$. We simplify the target as $\max_{\pi}Pr_{\pi}[\tau]   \phi^{\top}\Lambda^{-1}\phi$, which is actually $\sum_{(s,a,h,s',a',h')}   \mathbb{I}[(s_h,a_h)=(s,a), (s_{h'},a_{h'}) =(s',a')  ]W_{s,a,h,s',a',h'}$ where $W = \Lambda^{-1}$.
> We name it as the double-state reward function because the classical reward function could be viewed as a single-state reward function, where the reward only depends on one state-action pair. To optimize a double-reward function, the learner makes decisions according to its trajectory, which means its policy is non-Markovian. To our best of knowledge, there is no existing algorithms which could solve this problem efficiently, even if assuming the knowledge of the transition model.
>
>  **About $\tilde{\pi}$ in Algorithm 4, Line 7:** $\tilde{\pi}$ could be regarded as an approximate solution to Eq(6).
>   As discussed above, we do not know how to solve the optimization problem in line 4 algorithm 4 efficiently, which is the major difficult to compute $\tilde{\pi}$.

---

### Official Review · Reviewer_nsZ6 · 2024-11-01

**Soundness:** 3
**Presentation:** 2
**Contribution:** 3
**Rating:** 5
**Confidence:** 4

**Summary:**

The authors present a regret bound for RL with trajectory feedback. The lower bounds improves upon existing results reducing the dependency in the state space and are minimax optimal. To achieve the bounds, the authors use better confidence bounds than previous work and a more refined exploration scheme.

**Strengths:**

A strong theoretical result improving upon SOTA for an important problem

Refined techniques for a well studied problem

I appreciate the authors' attempt to provide an informal explanation of the proofs. It is not trivial.

**Weaknesses:**

I am curious to have a formal understanding of the term "for sufficiently large K" when presenting THM1. This is important because it can render the result.

The presentation of the algorithm is convoluted. Especially. it is not clear what is Design in Algorithm 5. In general, I think this can be grossly simplified.

The last term in THM 7 is kind of disappointing as it renders the result less exciting.

I believe the examples in the introduction were discussed by others before. A proper reference is needed.

There are some typos in the text that should be fixed: e.g., "ofte" (l46) and "fro" (l250)

**Questions:**

Q1: Please elucidate the initial K issue.

Q2: The paper seems incremental wrt Efroni et al (2021). What is the main contribution wrt Efroni et al?
Is it a mix and match of a model and know techniques? If not, are the new techniques applicable elsewhere?

Q3: Can you say something about the constant c>0 in l262? Does it depend on the problem itself?

Q4: L282-285: as far as I understand the statement "it holds that \pi^* \in \Pi_{\ell+1}" is a high probability even rather than an almost sure statement. So, if I don't understand what is going on. More specifically, for any finite T or K_\ell all elements in Eq (9) and after are random variables, but they seem to be treated as fixed because everything is taken in the limit. But it cannot be taken as such (at least not without a proof). I am probably missing something here, but this seems like a loose usage of \tilde{O} notations. Please explain.

Q5: How is K_0 set in Algorithm 1?

Q6: It seems to me that the computation needed for Algorithm 5, line 9 would be quite difficult. Is this correct? Can you say something about the complexity of the different algorithms?

Q7: What do you suspect is the real dependence of the low order terms in THM7?

I would be happy to raise my score if the questions above are answered properly, especially Q4.

---

> ### Author Response · Authors · 2024-11-22
> **Response to Reivewer nsZ6**
>
> **About burn-in time:**
> According to the final regret bound, the burn-in time would be $S^{24}A^{16}H^{28}$, which is impratically large. The main issue is from the initial step (line 3 in Algorithm 1) to learn the reference model, and it is a challenging problem to reduce this term (e.g., reduce the order of $H$ to tight).
>
> **About Design in Algorithm 5:**
> The Design step (Line 11-13, Algorithm 5) stems from experimental design, where the target is to find a distribution over a set of datapoints to maximize some certain potential function.
>
> **About the examples in the introduction:** Similar examples were indeed discussed by prior work on RL with trajectory feedback (e.g., Efroni et al (2021)) and we will add references in the next revision accordingly.
>
> **About comparison with (Efroni et al., 2021):** In result, the main contribution is a near-tight asymptotic regret bound. In technique, we formulate the RL problem as a linear bandit problem with feature as the trajectory, while (Efroni et al., 2021) used the occupancy distribution as features. The major difference is that, the number of all possible trajectories is $(SA)^H$, while the number of deterministic policies is $A^{SH}$. This is way we can save an $\sqrt{S}$ factor compared to the best result in Efroni et al., 2021.
>
> **About $c>0$ in line 262:** It does not depend on the problem. In fact, the exact inequality would be: with proability $1-\delta$,
> $\Lambda\geq \frac{1}{3}T\Lambda_{\bar{\pi}} -3\log(d/\delta)\mathbf{I}$.
>
>
> **About line 282-285:** We present (9) and (10) to explain the high-level idea. There is possibility that some of the concentration inequalities might fail, and the learner would suffer a linear regret $KH$. But this probability is very small, so we ignore this probability in explaining the intuitions. On the other hand, the exact definition of $\Pi_{\ell+1}$ is presented in Line 9 Algorithm 5, where we eliminated policies with value smaller than the best LCB.
>
>
> **About $K_0$:** We set $K_0 = 100000S^{4.5}A^{1.5}H^{8.5}\log(\frac{SAHK}{\delta})$ (see Appendix.A).
>
> **About computation cost  in Algorithm 5 line 9:** You are correct about this. In general this step is inefficient. But it is easy to test whether an element is in $\Pi_{\ell}$ since  $\max_{\pi}W^{\pi}(u,\hat{p})$ could be computed with optimistic backward planning, which means we can optimize some functions with good properties over$\Pi_{\ell}$ (e.g., value function). However,  the target function $\sum_{\tau}Pr_{\pi,p}[\tau]\min\\{ \phi_{\tau}\Lambda^{-1}\phi_{\tau} ,1\\}$ in Line 4 Algorithm 4   is more complicated, where we have to play enumeration to find the solution. This is the main difficulty in making the algorithm efficient in computation.

---

### Official Review · Reviewer_NMiJ · 2024-11-02

**Soundness:** 3
**Presentation:** 2
**Contribution:** 2
**Rating:** 5
**Confidence:** 3

**Summary:**

This paper considers the episodic reinforcement learning with trajectory feedback. In this paper, the authors proposed an online learning algorithm that achieves optimal regret which scales as sqrt(SAH^3K). Unlike existing methods directly applied linear bandit, the proposed algorithm does not suffer from the regret with higher order on S.

**Strengths:**

This paper provides rigorous theoretically analysis for the proposed approach. Instead of building confidence region as linear bandit, the proposed algorithm maintain a policy set within a constant range, which can be done through data sampling in batches. The theoretical motivation behind this idea is very clearly presented in the paper with rigorous proofs.
Comparing to existing work, the proposed algorithm achieve optimal regret bound for RL with trajectory feedback only. Given the trajectory feedback is more general for real application, the proposed work has a potential to be applied to real problem with better performance guarantees.

**Weaknesses:**

The paper focuses on proposing algorithm that has tighter regret bound. While the rigorous proofs are appreciated, a more comprehensive comparison would help with understanding of the importance of the difference. It will become clearer with some toy experiment showing how the linear bandit performance in the episodic MDP environment when state/action is huge and time horizon is long.
And since the proposed algorithm still require exponential time cost, it is also needed to compare the complexity in the paper.

**Questions:**

Beside what I wrote in the previous section, I would also like to understand more about Theorem 1 and 7. The statement of 'a RL problem with trajectory feedback has the same asymptotically optimal regret as standard RL' is a very intriguing result. I would like to read more explanation on the interpretation of Theorem 7 in the papaer.

---

> ### Author Response · Authors · 2024-11-22
> **Response to Reviewer NMiJ**
>
> Please find our answers to the questions about **inefficiency** and  **why trajectory-feedback RL is as easy as standard RL** in the common response.

---

### Official Review · Reviewer_UYMk · 2024-11-03

**Soundness:** 3
**Presentation:** 3
**Contribution:** 3
**Rating:** 8
**Confidence:** 2

**Summary:**

The paper studies a RL problem with delayed rewards, which are sampled after letting a trajectory roll out for a certain time horizon. This is a practical scenario that appears in several applications and that, to date, doesn't have a detailed and comprehensive solution. The paper shows how optimal regrets can be obtained that are similar to those of traditional RL algorithms.

**Strengths:**

The paper studies a novel and interesting variant of the standard RL problem, where rewards are generated with a delay rather than at each time instants. The provided bounds are technically sound and improve upon the existing literature.

**Weaknesses:**

1. Rewards are assumed to be a linear function of the trajectory (sequence of states and actions). This may not be practical in real-life. This is not practical even in the case discussed in the introduction - medical care.

2. Learning the optimal policy, with the assumption of linear rewards, is now a (linear) regression problem. This is the main part from where the similarity between regret bounds for traditional RL and the proposed method arises. Can you comment on more complicated or realistic bandit models?

3. Using a parameterized policy that can be systematically updated after the end of each trajectory, instead of policy elimination, might result in tighter regret bounds. Is this the case?

**Questions:**

Do the methods/ideas extend to cases where rewards are nonlinear functions of the trajectories?

The connections to linear bandits is interesting, and it would be beneficial to expand upon this connection (formulation and examples). The literature in this area is also rather extensive, and a more comprehensive review/comparison would be helpful.

Some numerical/illustrative examples could help the reader appreciate the theoretical results.

---

> ### Author Response · Authors · 2024-11-22
> **Response to Reviewer UYMk**
>
> **About more complicated or realistic bandit models:** In this work, we introduce linear bandit problem due to the natural formulation of reinforcement learning. As for extensions, it would be an interesting problem to replace the tabular MDP with more complicated MDPs, e.g., (generalized) linear MDP. We think advanced bandit models are related to these problems.
>
> **About parameterized policy without elimination:** As far as we can see, it is unclear that parameterized policies could help improve the regret bound. On the other hand, we make delayed updates to keep statistical independence, so it might be improper to make frequently updates.
>
>
> **About the case where ewards are nonlinear functions of the trajectories:** The general idea is to formulate the problem as a bandit problem with structure. There is possibly more interesting formulation because the arm set in this problem is the set of trajectories, which also has a good structure.
>
> **About connections to linear bandits:** Thanks for the suggestion and we will improve the presentation accordingly.

---

### Official Review · Reviewer_zmph · 2024-11-04

**Soundness:** 3
**Presentation:** 3
**Contribution:** 3
**Rating:** 6
**Confidence:** 2

**Summary:**

This paper addresses reinforcement learning with trajectory feedback, where the reward in a Markov Decision Process is not observed throughout an episode but only as a cumulative reward at the episode’s end. The authors propose an algorithm that achieves a regret bound with statistical behavior comparable to the state of the art in standard reinforcement learning literature.

**Strengths:**

The paper proposes an algorithm that achieves a better asymptotic regret bound than those previously established for reinforcement learning with trajectory feedback. The authors employ a novel method in their proofs, which appears robust.

**Weaknesses:**

First, I must admit that I am not well-versed in the literature on regret bounds in reinforcement learning with bandits.

To me, the framework of this paper appears quite limited. Moreover, the asymptotic regret bounds seem similar to those achieved in more classical frameworks, giving the impression that the work involves only minor modifications to standard proofs. However, I may be mistaken, as I am not deeply familiar with this literature.

**Questions:**

Here are few questions :
1. Could the authors elaborate on the statement in lines 261–262: "Therefore, by running [...] for some constant $c>0$"? A more detailed explanation of this step would be helpful.
2. The definition of the set in (10) seems unclear to me. First, how do the authors compute $\max W^{\pi'}(\hat{R},P)$? Additionally, how is the quantity on the right-hand side of the selection criterion, i.e., the practical value of $\tilde{O}$?
3. Can the method employed in this paper be adapted to consider cases where $K$ is not that large compared to $S$, $A$ and $H$, similar to the approach in [Zhang et al.]?

---

> ### Author Response · Authors · 2024-11-22
> **Response to Reviewer zmph**
>
> **About the constant $c$ in lines 261-262:** By this statement, we mean that by running $\pi$ for $T$ rounds to
> get trajectories
> $\tau_i$
> with feature $\phi_{\tau_i}$ for $1\leq i \leq T$, the information matrix $\Lambda = \sum_{i=1}^{T}\phi_{\tau,i}\phi_{\tau,i}^{\top}$ would be an upper bound of $\frac{1}{2} T E_{\pi}[\phi_{\tau}\phi_{\tau}^{\top} ]$ with high probability (allowing some logarithmic error terms).
>
> **About equation (10):** Our goal was to present some high-level intuition. In the algorithm, we do not compute $\Pi_{\ell+1}$ with (10). Instead,
> we eliminate policies by comparing the value of some policy with the best LCB (lower confidence bound) over all policies (see Line 9 in Algorithm 4). By presenting equation (10), we aim to show that, for any policy $\pi\in \Pi_{\ell}$ the gap between its UCB and its LCB is at most $\tilde{O}(\sqrt{SAH^3/K_{\ell}})$. To compute $\max_{\pi}W^{\pi}(\hat{R},P)$, we can run simple backward planning since here we assume the knowledge of $P$. In the case $P$ is unknown, we compute $\max_{\pi}W^{\pi}(\hat{R},\hat{P})$ with optimistic backward planning, where $\hat{P}$ is the empirical transition model.
>
>
>  **About small $K$:** As far as we see, the answer is negative. Our algorithm depends on the knowledge of a reference transition model which requires a large sample complexity (in $S$, $A$ and $H$) using current techniques.

---

### Meta-Review · Area_Chair_GCCv · 2024-12-17

**Metareview:**

This work contributes a new algorithm for online RL with a special form of information feedback. However, the manuscript itself is full of typos, the importance of the results is not demonstrated experimentally, and there do not appear to be any highly innovative components in the planning and policy learning steps. For this reason, the paper does not meet the bar for acceptance at this time.

**Additional Comments On Reviewer Discussion:**

NA

---

### Decision · Program_Chairs · 2025-01-22

Reject